# A SYSTEMATIC STUDY ON EARLY STOPPING CRITERIA IN HPO AND THE IMPLICATIONS OF UNCERTAINTY

## ABSTRACT

The development of hyperparameter optimization (HPO) algorithms constitutes a key concern within the machine learning domain. While numerous strategies employing early stopping mechanisms have been proposed to bolster HPO efficiency, there remains a notable deficiency in understanding how the selection of early stopping criteria influences the reliability of early stopping decisions and, by extension, the broader outcomes of HPO endeavors. This paper undertakes a systematic exploration of the impact of metric selection on the effectiveness of early stopping-based HPO. Specifically, we introduce a set of metrics that incorporate uncertainty and highlight their practical significance in enhancing the reliability of early stopping decisions. Through a series of empirical experiments conducted on HPO and NAS benchmarks, we substantiate the critical role of metric selection, while shedding light on the potential implications of integrating uncertainty as a criterion. This research furnishes empirical insights that serve as a compass for the selection and formulation of criteria, thereby contributing to a more profound comprehension of mechanisms underpinning early stopping-based HPO.

## 1 INTRODUCTION

Hyperparameter Optimization (HPO) plays an important role in machine learning (ML) (Bischl et al., 2023; Yang & Shami, 2020; Feurer & Hutter, 2019). The selection of hyperparameters, such as learning rate, batch size, and network architecture, is essential for models' performance. HPO is time-consuming as it typically involves the trainings of many candidate ML models with different hyperparameters (Wu et al., 2019; Turner et al., 2021). *Early stopping* is a method commonly used in HPO to make HPO fit into an acceptable time budget (Akiba et al., 2019; Nguyen et al., 2020; Makarova et al., 2022). It achieves that by halting the trainings of certain candidate models before their convergence if they are considered not promising by some *early stopping metrics*. In an HPO scheme (e.g., Hyperband (Li et al., 2017)), early stopping is often applied repeatedly throughout the HPO process such that more unpromising models are terminated as the process goes. These methods, also known as multi-fidelity optimization algorithms, introduce varying levels of resources or computational costs into the optimization process (Falkner et al., 2018; Awad et al., 2021; Swersky et al., 2013; Wu et al., 2020; Takeno et al., 2020). Early stopping is essential for those methods to strike a good balance between HPO cost and the effectiveness of the final models (Forrester et al., 2007; Klein et al., 2017; Kandasamy et al., 2017; Li et al., 2017).

While prior studies have explored various early stopping-based HPO algorithms (Eggensperger et al., 2021; Bansal et al., 2022; Wistuba et al., 2022; Yan et al., 2021; Meng et al., 2021), early stopping criterion, particularly the performance metrics used therein for model ranking, remains preliminary understood. Early stopping metrics directly determines what models to keep and what models to discard. Despite its pivotal role for HPO, no prior studies have systematically explored it. Existing HPOs have been using either training loss/accuracy or validation loss/accuracy as the early stopping metric; which one to use is based on the practitioners' personal preferences, with validation loss being a more frequent choice.

Some fundamental questions on early stopping metric remain open:

(1) How reliable are the commonly used performance metrics for HPO? How do they compare to one another?
(2) How to explain the reasons for the different effectiveness of the metrics? More fundamentally, what are the nature of early stopping and the key factors for its effectiveness?
(3) Besides the commonly considered measures, are there any other measures worth considering for early stopping metrics? More specifically, ML models have inherent uncertainty. How would model uncertainty impact early stopping? Would it be worthy to incorporate it into early stopping metrics for HPO?

This paper aims to answer these fundamental questions, and advance the principled understanding of early stopping in HPO. We do that through a three-fold exploration. (i) We first conduct an empirical study on nine HPO tasks in two widely used HPO benchmarks (Nas-Bench-201 and LCBench) over nine datasets, and systematically examine the effectiveness of the popular early stopping metrics for HPO. We use the concept of reliability to assess if a specific performance metric shows statistically significant superiority over others. (ii) From the data, we distill a set of insights on the relative effectiveness of the popular early stopping metrics, theoretically analyze the inherent nature of early stopping, and reveal the reasons for the pros and cons of those metrics. (iii) We further study the impact of model uncertainty - the variations in model predictions - on early stopping, propose a set of metrics that integrate model uncertainty, and uncover the potential of incorporating model uncertainty into early stopping metrics for HPO.

To the best of our knowledge, this work is the first that gives a systematic exploration on early stopping metric for HPO. By addressing some fundamental open questions, it advances the understanding of early stopping for HPO, provides novel insights and empirical guidelines for metric design and selection, and paves the way for further advancing the field of HPO.

In the rest of the paper, we will first report our empirical studies on the effectiveness of the popular early stopping metrics (Section 2), and then provide our theoretical analysis of early stopping and the reasons for the observed results (Section 3). We finally report our investigations of the impact from model uncertainty and some possible ways to incorporate model uncertainty into early stopping metrics (Section 4).

The main observation of this paper is as follows:

> (i) Different metrics may have significantly different impacts on the HPO performance, and training loss tends to suit complex tasks better than validation loss.
> (ii) The risk bound of early stopping can be well formulated, and metrics on large datasets with a more discerning capability have a lower risk bound.
> (iii) Model uncertainty has some important impact on the effectiveness of early stopping.
> (iv) Careful combination of model uncertainty with conventional early stopping metrics can yield significant benefits for HPO, especially when dynamic trends of the training process are considered.

## 2 EMPIRICAL STUDY ON EARLY STOPPING METRICS FOR HPO

In early stopping-based HPO, performance metrics play a critical role in assessing each configuration's capability at a specific fidelity level, informing the early stopping decision for filtering. An effective metric should consistently reflect a model's present performance and its potential for improvement. Despite extensive research on early stopping-based HPO methods, there is a notable absence of consensus regarding the selection and reliability of metrics for guiding early stopping decisions. In this section, we address this gap by empirically comparing commonly used metrics to unveil underlying differences. We start by detailing the experimental setup applied across all experiments throughout this paper.

### 2.1 EXPERIMENTAL SETUP

**Methodology.** We assess the reliability of common metrics within single-objective classification tasks using Hyperband, BOHB, and Sub-sampling (SS) algorithms. The results on Hyperband, representing our primary findings, are presented in the main text. Additional details on Hyperband, BOHB, and SS can be found in Appendix A, and their respective results are elaborated in Appendices B and E.

The early stopping metrics we examine include training accuracy, training loss, validation accuracy, and validation loss. To validate the reliability of these metrics, we conduct experiments across various benchmarks, budget constraints ($R$), and filtering ratios ($\eta$) within the Hyperband algorithm, as described in Appendix A. For each setting, we perform 1000 repetitions with different random seeds, each including a randomly selected subset of model configurations. We compare the outcomes of early stopping decisions guided by different metrics, and we employ the Wilcoxon signed-rank test (Woolson, 2007) to determine the presence of significant differences among the metrics. For significantly different groups, we quantify the difference between their group means by using standardized effect size called Cohen's d (Cohen, 2013), as defined in Eq. A.1. To report results, we consider indicators such as *final performance*, *performance over time*, and *performance regret* (defined as the discrepancy between the best-found value and the best-known value) (Eggensperger

et al., 2021). Additionally, we count the number of wins, ties, and losses of each metric across the 1000 repetitions.

**Benchmarks.** To ensure the broad applicability of our findings, we undertake comprehensive testing across two widely recognized benchmarks of varying scales: LCBench (Zimmer et al., 2021) and Nas-Bench-201 (Dong & Yang, 2020). LCBench comprises six lightweight HPO tasks, drawn from MLPs trained on established OpenML datasets such as Fashion-MNIST, jasmine, and vehicle. Nas-Bench-201 encompasses three heavyweight NAS tasks, derived from cell-based architectures trained on CIFAR-10, CIFAR-100, and ImageNet-16-120 datasets. Details related to these benchmarks are provided in Appendix A. In subsequent discussions, we utilize the dataset names to represent each respective benchmark.

## 2.2 OBSERVED RELIABILITY OF COMMON METRICS

We present in Figure 1 the average test accuracy (top row) achieved using diverse metrics across various budget constraints, along with the average regret (bottom row) of the remaining model configurations subsequent to decisions made by different metrics during the HPO progresses. Opposite to the common inclinations towards the use of validation metrics, in Nas-Bench-201, characterized by high task complexity, the use of training metrics consistently outperforms the use of validation metrics across all budget constraints, exhibiting an average difference of 0.72% and a maximum difference of 6.54% in test accuracy. Conversely, in LCBench, where complexity is lower, validation metrics tend to outperform training metrics in most scenarios, with an average difference of 0.39% and a maximum difference of 10.64%. Furthermore, we observe that the disparity between early stopping decisions made by different metrics increases as model training progresses. In Nas-Bench-201, training metrics, particularly training loss, exhibit a preference for retaining superior model configurations. Additionally, we investigate the impact of budget constraints, noting that in LCBench, exemplified by the jasmine benchmark, higher budget constraints tend to have a detrimental effect on the efficacy of training metrics.

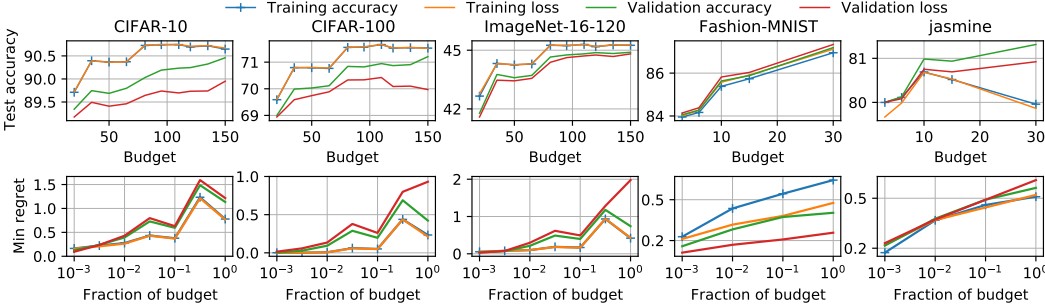

Figure 1: Mean accuracy across 1000 repetitions of optimal configurations selected with commonly used early stopping metrics under diverse budget constraints (upper row). Mean optimal *regret-over-time* filtered with commonly used metrics (lower row). "Fraction of budget" denotes the proportion of allocated budget used during training. See Appendix B.1 for other datasets.

In conjunction with qualitative assessments, we employ a sign test to quantitatively measure the extent of variability among different metrics. We tally their respective wins, ties, and losses, presenting the detailed results in Table 1 (comprehensive results are available in Appendix B). Notably, significant disparities are evident among nearly all pairs of metrics, with training loss demonstrating a higher likelihood of success in complex tasks, while validation loss prevails in simpler tasks.

Table 1: P-value from a Wilcoxon signed-rank test for the hypothesis that *training loss* and *validation loss* outperform commonly used metrics. We also give the number of wins and losses of each metric against *training loss* and *validation loss* over 1000 repetitions.

| | CIFAR-10 | | | CIFAR-100 | | | ImageNet-16-120 | | |
|---|---|---|---|---|---|---|---|---|---|
| Against train. loss | Train. acc. | Valid. loss | Valid. acc. | Train. acc. | Valid. loss | Valid. acc. | Train. acc. | Valid. loss | Valid. acc. |
| P-values | 0.04 | $7.1e^{-133}$ | $5.4e^{-106}$ | 0.17 | $1.1e^{-98}$ | $2.9e^{-66}$ | 0.18 | $1.4e^{-66}$ | $1.2e^{-46}$ |
| Wins/losses | 46/75 | 156/631 | 225/447 | 30/31 | 120/770 | 209/403 | 24/30 | 146/382 | 143/338 |

| | Fashion-MNIST | | | jasmine | | | vehicle | | |
|---|---|---|---|---|---|---|---|---|---|
| Against valid. loss | Valid. acc. | Train. loss | Train. acc. | Valid. acc. | Train. loss | Train. acc. | Valid. acc. | Train. loss | Train. acc |
| P-values | $4.7e^{-22}$ | $2.4e^{-14}$ | $3.5e^{-32}$ | 0.99 | 0.03 | 0.008 | $2.1e^{-4}$ | 0.44 | $2.1e^{-6}$ |
| Wins/losses | 39/139 | 76/241 | 69/315 | 267/160 | 164/643 | 176/604 | 248/140 | 54/77 | 133/231 |

We distill these observations into the following insight:

*Insight 1: For complex tasks (which take many epochs to converge), training metrics outperform validation metrics in serving for early stopping in HPO.*

## 3 THE REASONS AND THEORETICAL ANALYSIS

This section examines the reasons for the observed differences in the effectiveness of the common metrics, and more importantly, reveals the underlying factors and how they influence the effectiveness of early stopping metrics.

Before delving into a detailed analysis, it is worthwhile to first examine the models' performance throughout the training lifecycles. In Figure 2, we present the *performance-over-time* curves for validation and training losses on the ImageNet-16-120 and Fashion-MNIST benchmarks. In the case of the complex ImageNet-16-120 benchmark, the validation loss curve displays considerable volatility, whereas the training loss follows a smoother and more consistent trajectory. In contrast, on the lightweight Fashion-MNIST dataset, model configurations converge in only several epochs. As a result, early stopping decisions guided by training metrics beyond this point may select model configurations that already overfit. This observation sheds light on the reason why early stopping decisions based on training metrics tend to yield sub-optimal results on LCBench as budget constraints become more lenient, contributing to the improved variability in outcomes. Next, we undertake a theoretical analysis to uncover its underlying causes.

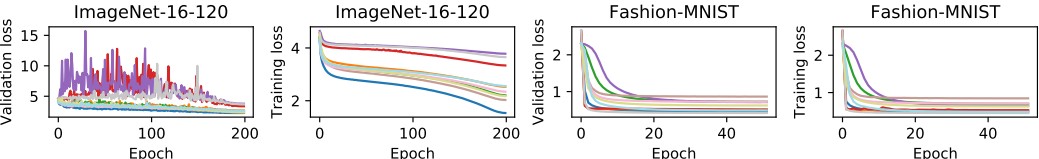

Figure 2: Evolution of validation and training losses across epochs for 10 random configurations.

We start by analyzing the probability for an early stopping strategy to make a sub-optimal decision before convergence. Consider an HPO task under a supervised learning scenario where a machine learning model $M$ is trained on a set of data points $D_T = \{(x_i, y_i)\}_{i=1}^n$, sampled i.i.d. from some unknown data distribution $U$. Let there be $K$ hyperparameter candidates $\gamma_1, \gamma_2, \dots, \gamma_K \in \Gamma$. We denote the model trained with hyperparameter $\gamma$ for $t$ epochs as $M_\gamma^t$ and the converged model as $M_\gamma^*$. Given some loss function $\ell(\cdot, \cdot) \in [lb, ub]$ ($lb \geq 0$), the *expected risks* of models $M_\gamma^t$ and $M_\gamma^*$ with respect to $U$ are respectively defined as:

$$f^t(\gamma) = \mathbb{E}_U\left[\ell\left(\boldsymbol{y}, M_\gamma^t(\boldsymbol{x})\right)\right], \quad \text{and} \quad f^*(\gamma) = \mathbb{E}_U\left[\ell\left(\boldsymbol{y}, M_\gamma^*(\boldsymbol{x})\right)\right]. \tag{3.1}$$

The main objective of HPO is to identify hyperparameters $\gamma_o$ that minimize the expected risk of converged models, expressed as: $\gamma_o = \arg\min_{\gamma \in \Gamma} f^*(\gamma)$. However, in practice, the expected risk cannot be directly computed as $U$ is unknown. Consequently, it relies on the estimation of the expected risk using a finite set $D$ drawn i.i.d. from the distribution $U$. Thus, practical HPO centers around minimizing the *empirical estimate*:

$$\hat{f}^*(\gamma) = \frac{1}{|D|} \sum_{x_i, y_i \in D} \ell\left(y_i, M_\gamma^*(x_i)\right). \tag{3.2}$$

**Proposition 1.** *Consider an early stopping-based HPO that employs model's loss function as its early stopping metric. Let $f^t$ and $\hat{f}^t$ denote the expected and empirical losses at any epoch $t$ before convergence, and let $f^*$ and $\hat{f}^*$ denote the expected and empirical losses at the convergence time, as defined in Eqs. 3.1 and 3.2. Assume that $f^t(\gamma) \geq f^*(\gamma)$ and $\hat{f}^t(\gamma) \geq \hat{f}^*(\gamma)$ hold for all $\gamma \in \Gamma$. Let $\gamma_o = \arg\min_{\gamma \in \Gamma} f^*(\gamma)$ be the optimal hyperparameter, and let $\gamma_{so}$ denote a candidate solution with sub-optimality. Then the probability of making an incorrect early stopping decision at epoch $t$ can be bounded according to Markov's inequality:*

$$P\left(\hat{f}^t(\gamma_o) - \mathbb{E}\hat{f}^t(\gamma_{so}) \geq 0\right) \leq \frac{\mathbb{E}\hat{f}^t(\gamma_o)}{\mathbb{E}\hat{f}^t(\gamma_{so})} = \frac{f^t(\gamma_o)}{f^t(\gamma_{so})}. \tag{3.3}$$

*Based on Hoeffding's inequality Hoeffding (1994) and the relations between $f^t(\gamma_{so})$ and $f^t(\gamma_o)$, we derive tighter bounds (see Appendix C for proof):*

$$P\left(\hat{f}^t(\gamma_o) - \mathbb{E}\hat{f}^t(\gamma_{so}) \geq 0\right) \leq e^{-\frac{2|D|\left(f^t(\gamma_{so}) - f^t(\gamma_o)\right)^2}{(ub-lb)^2}}, \quad \text{if } f^t(\gamma_{so}) > f^t(\gamma_o)$$

$$P\left(\hat{f}^t(\gamma_o) - \mathbb{E}\hat{f}^t(\gamma_{so}) \geq 0\right) \geq 1 - e^{-\frac{2|D|\left(f^t(\gamma_{so}) - f^t(\gamma_o)\right)^2}{(ub-lb)^2}}, \quad \text{if } f^t(\gamma_{so}) \leq f^t(\gamma_o).$$

$$\tag{3.4}$$

This proposition establishes a theoretical basis for bounding the probability of early stopping decision errors in terms of both the dataset size $|D|$ and expected loss variance. The larger $|D|$ or $\left(f^t(\gamma_{so}) - f^t(\gamma_o)\right)^2$ is, the smaller the risk bound is. It suggests preference for metrics that are built on larger datasets and metrics that lead to larger gaps between models at epoch $t$. It is intuitive: larger datasets lead to reduced discrepancies between estimates and expectations, increasing the likelihood of obtaining precise measurements; metrics that effectively differentiate among various models are more likely to single out the inferior models. These factors collectively explain why, in the context of Nas-Bench-201, training loss demonstrates higher reliability as an early stopping metric compared to validation loss. It suggests that while a larger training dataset may contribute, the primary justification lies in the training loss serving as a superior indicator of model expressiveness, yielding greater stability and consistent model ranking. In contrast, the validation loss prioritizes the aspect of generalization and tends to favor models with lower capacities that converge early. But for simple models that converge fast, in the stage after overfitting, training loss fails to discern $f^t(\gamma_{so})$ and $f^t(\gamma_o)$; in light of the proposition, in that case, the training losses of sub-optimal models may be lower than those of optimal models, contradicting the condition of the proposition that $f^t(\gamma) \geq \hat{f}^*(\gamma)$. It is hence important to discern the cases of overfitting from the general cases, as the assumptions and implications differ. Note that our discussion pertains to a specific context of "one early-stopping based HPO", indicating a consistent DL task, comparable model complexities (varied hyperparameters), and fixed training, validation, and testing datasets.

From the discussion, we distill the following insight:

***Insight 2: metrics that are based on larger datasets and lead to larger gaps in metric values between different models tend to be more helpful for early stopping in HPO.***

Based on our findings, several strategies exist to enhance the reliability of the early stopping approach. Ideally, we could seek a metric that allows more robust models to consistently exhibit superior values at any epoch $t$. However, achieving such a metric is highly challenging, often demanding significant computational resources to comprehensively understand the behavior of all model configurations in advance. An alternative approach is to maximize the model's expressive capacity based on its available information, thereby generating robust and superior metrics that can serve as a dependable basis for early stopping. On one hand, if we have prior knowledge about the model's convergence behavior, one effective strategy involves integrating both training and validation losses. This integration allows the model to balance the expressiveness exhibited in the training loss against the model's generalization ability as indicated by the validation loss. We conduct some experiments, reported in Appendix D. On the other hand, by drawing insights from the model's behavior in terms of training and validation profiling, as illustrated in Figure 2, the incorporation of model uncertainty into the metric is a viable strategy. This uncertainty encapsulates both the potential expressiveness of the model and its inherent instability. Effectively utilizing uncertainty can establish a more robust foundation for early stopping. We next delve into a more detailed exploration of the implications and effects of uncertainty.

## 4 Model Uncertainty and Early Stopping in HPO

This section gives the first known exploration of the effects of model uncertainty on early stopping and ways to incorporate it into early stopping in HPO. Uncertainty, embodying the inherent variability in model predictions, represents a double-edged sword in this context. On one hand, it introduces instability to the model and poses challenges in demonstrating the model's true capabilities. On the other hand, it provides valuable insights into the latent potential and capabilities of the model. We next delve into the manifestations and impacts of model uncertainty, and explore how leveraging uncertainty can guide the formulation of more reliable early stopping decisions.

### 4.1 Manifestations of uncertainty

Uncertainty in machine learning arises from two primary sources: intrinsic noise within the data and variability in model predictions stemming from limited knowledge (Hüllermeier & Waegeman, 2021; Abdar et al., 2021). While data uncertainty remains a constant, it is the variability in model predictions, referred to as model uncertainty, that predominantly affects early stopping decisions.

Building upon the research conducted by Zhou et al. (2022) regarding uncertainty in machine learning, and in alignment with our definitions in Section 3, we assume that the model predictions adhere to a probability distribution, $\hat{M}_\gamma^t(\boldsymbol{x}) \sim \mathcal{N}(\hat{\boldsymbol{y}}, \sigma_t^2)$, where the distribution centers around the mean value $\hat{\boldsymbol{y}} = \mathbb{E}\hat{M}_\gamma^t(\boldsymbol{x})$. Let $\boldsymbol{y} = g(\boldsymbol{x}) + \varepsilon$ be an observation corresponding to a given input

$\boldsymbol{x} \in \mathbb{R}^d$, perturbed by noise $\varepsilon \sim \mathcal{N}(0, \sigma^2)$. Therefore, $\boldsymbol{y} \sim \mathcal{N}(g(\boldsymbol{x}), \sigma^2)$, where $g(\boldsymbol{x})$ represents the ground truth, and $\sigma$ represents the noise arising from data, which is fundamentally irreducible.

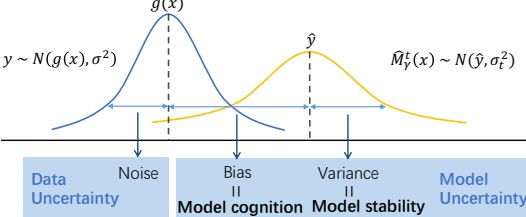

Figure 3 shows the probability distribution of observations and model predictions for a specific $\boldsymbol{x}$ at epoch $t$, illustrating the concept of uncertainty decomposition. The "*Bias*", quantifying the gap between $\hat{\boldsymbol{y}}$ and $g(\boldsymbol{x})$, denotes the difference between estimated and true values. It reflects the model's cognitive capacity considering various training configurations such as hyperparameters, fidelity, and learning algorithms. On the other

Figure 3: Decomposition of uncertainty.

hand, "*Variation*" is linked to the model's sensitivity to the training samples. Notably, when computing the empirical loss of the learned model at epoch $t$, this value encompasses not only the model's current cognitive level but also the uncertainty linked to model stability. Accordingly, optimal early stopping decisions should primarily rely on the model's current cognitive capabilities.

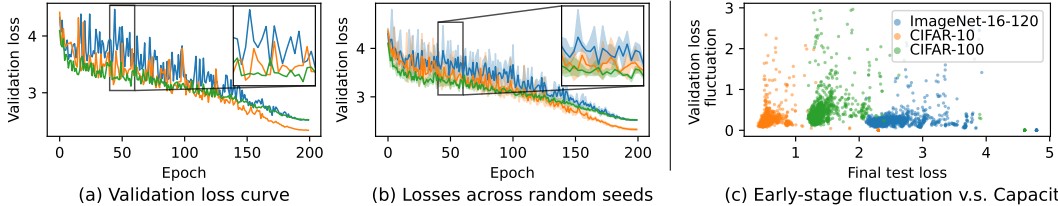

| (a) Validation loss curve | (b) Losses across random seeds | (c) Early-stage fluctuation v.s. Capacit |

Figure 4: Manifestation of model uncertainty. (a) and (b) showcase three randomly selected models from the ImageNet benchmark.

The influence of variation on the model's cognition manifests in two notable aspects: Firstly, model uncertainty is observable through significant fluctuations in the prediction occurring over a limited range of epochs. This phenomenon is particularly evident in the validation set when the model has not yet fully captured the underlying data patterns. This fluctuation is exemplified by the validation loss curves depicted in Figure 4 (a), where pronounced oscillations occur during the unconverged phase, typically within the first 100 epochs. These fluctuations tend to diminish as the model approaches convergence (epochs exceeding 150). An intriguing observation, as illustrated in Figure 4 (c), reveals a noteworthy pattern: models with greater expressive capacity tend to exhibit higher levels of early-stage fluctuations. This stems from the fact that stronger models harbor increased uncertainty during their initial phases when their knowledge has not yet aligned with their expressive capabilities. In contrast, models with limited expressiveness tend to converge earlier and display reduced uncertainty. Consequently, if an early stopping decision is made at a point of peak uncertainty, stronger models may be prematurely terminated. Secondly, model behaviors are notably influenced by various training settings, including aspects such as initialization and the sequence of data batch loading. These diverse training settings introduce variations in the model's learning process, enabling it to capture distinct facets of the data, thereby leading to varying biases and error patterns. Figure 4 (b) illustrates the mean and variance of validation losses across three random seeds, illustrating the impact of training settings on model performance. This variability, when effectively harnessed, will provide a holistic view of the model's cognitive capabilities. It is hence helpful to reduce the influence of variance in early stopping metrics to provide a more precise representation of the model's competence for HPO.

## 4.2 IMPLICATIONS OF UNCERTAINTY INTEGRATION

Next, we explore strategies for mitigating the misleading aspects of uncertainty that appear as noise, and for harnessing the informative aspects of uncertainty that reveal the model's potential to aid in making informed early stopping decisions. For that, we propose and experiment with a set of metrics that incorporate uncertainty in different ways.

### 4.2.1 RELIABLE MODEL COGNITION

We begin by examining metrics that provide a robust characterization of a model's cognitive capabilities. This ensures a reduction in the risk of sub-optimal early stopping decisions that may arise due to random fluctuations. As depicted in Figure 4 (a) and (b), uncertainty can manifest as performance fluctuations across epochs and optimization processes. Distinguishing whether these variations signify actual model improvements or merely result from random noise can be challenging. To counteract the influence of these transient fluctuations and enhance the stability of early stopping

decisions, we first experiment with two metrics: *performance smoothing* and *ensemble averaging*. See Appendix E.1 for more technical details.

▶ **Performance smoothing.** Performance smoothing operates under the premise that performance fluctuations observed between neighboring epochs primarily stem from uncertainty rather than significant model improvements or deterioration. This method involves calculating the average performance over a small window of neighboring epochs, formally expressed as $\hat{C}_\mu^t(\gamma) = \frac{1}{|W|} \sum_{i=1}^{|W|} \hat{f}^{t-i+1}(\gamma)$, where $|W|$ denotes window size, and $\hat{f}^{t-i+1}(\gamma)$ represents the empirical loss at epoch $(t-i+1)$ for a specific hyperparameter configuration $\gamma$. This metric aims to provide a stable representation of the model's performance during training by averaging out short-term, stochastic fluctuations.

▶ **Ensemble averaging.** In an effort to further mitigate the impact of these variations, we explore the widely adopted ensemble learning metrics (Nixon et al., 2020; Fort et al., 2019; Ganaie et al., 2022a; Rahaman et al., 2021; Ganaie et al., 2022b). An ensemble is composed of independently trained models, all sharing the same configuration but operating on resampled datasets with differing initializations (Ovadia et al., 2019). The ensemble averaging metric calculates the average predictions generated by the ensemble of models at a specific epoch $t$, formally expressed as: $\hat{C}_{en}^t(\gamma) = \frac{1}{|E|} \sum_{i=1}^{|E|} \hat{f}_i^t(\gamma)$, where $|E|$ denotes the size of the ensemble, and $\hat{f}_i^t(\gamma)$ represents the empirical loss of the $i^{th}$ configuration within the ensemble. The ensemble averaging approach aims to provide a more stable and reliable measure of the model's performance by leveraging the wisdom of multiple optimization processes.

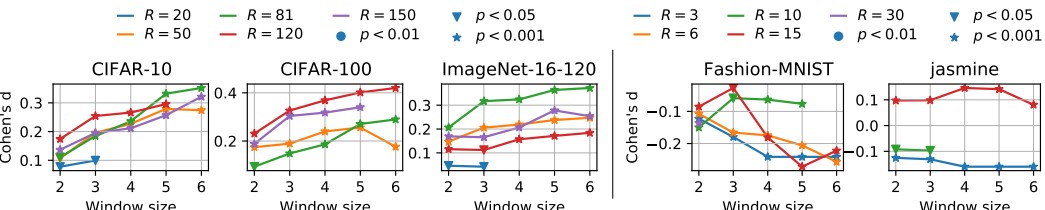

Figure 5: Effect size across 1000 repetitions of performance smoothing metrics over empirical losses. $R$ denotes the budget constraint of Hyperband, and $p$ represents the p-value from the Wilcoxon test.

The performance smoothing metric, as a stabilization strategy, offers a robust control over epoch-wise results. Its objective is to ensure the reliability of performance across epochs, mitigating the influence of abrupt performance spikes at specific time points. Figure 5 illustrates the impact of applying performance smoothing to empirical validation losses. The experiment includes a range of smoothing window sizes (2 to 6) to assess potential enhancements across various task complexities. The findings in Figure 5 unveil the performance smoothing settings that lead to statistically significant performance gains or losses, determined through the Wilcoxon test, along with the corresponding Cohen's d effect sizes, as defined in Eq. A.1. Positive Cohen's d values signify an augmented reliability attributed to the performance smoothing metrics. Notably, the results indicate the following findings. First, performance smoothing consistently leads to improved results, particularly in complex tasks characterized by extended convergence times. The most significant enhancements are observed when we use window sizes around 5 or 6, yielding effect sizes of approximately 0.3. Second, for tasks with lower complexity, opting for a smaller window size may be more appropriate. Rapid changes in empirical losses for these tasks suggest that excessive smoothing could potentially lead to a loss of genuine model performance. Third, when applied to training metrics, the impact of performance smoothing is less pronounced due to the inherently weak fluctuations in the training loss curve. Comprehensive details of all performance smoothing results are provided in Appendix E.2. These results underscore the effectiveness of performance smoothing metrics in mitigating stochastic fluctuations arising from overly biased results, thereby significantly enhancing the overall reliability of HPO based on early stopping.

We present in Figure 6 the outcomes of applying ensemble averaging to commonly used metrics. In this representation, darker colors correspond to the HPO processes employing the commonly used metrics, while lighter shades represent processes incorporating the ensemble averaging metrics. The figure highlights the significant distinctions introduced by the ensemble averaging metrics when contrasted with the original empirical metrics. This underscores the efficacy of the ensemble averaging metrics in mitigating uncertainty and enhancing the dependability of the early stopping

Figure 6: Performance across 1000 repetitions of ensemble averaging metrics over empirical losses. *=p-value<0.05, **=p-value<0.01, ***=p-value<0.001.

strategy. In most instances, the ensemble averaging metrics yield substantial improvements, with more pronounced effects observed on the validation set. Ensemble averaging adds significant computational burdens due to the necessity of repeatedly training models. We use it to primarily examine the potential of uncertainty-awareness for early stopping rather than advocate it as a solution. Next subsection discusses more practical ways to leverage model uncertainty.

The analysis leads to the following insight:

***Insight 3: Mitigating misleading uncertainty can help increase the reliability of early stopping metrics for HPO.***

### 4.2.2 POTENTIAL EXPLOITATION

In Section 4.2.1, we discussed the reduction of misleading aspects of uncertainty that manifest as noise to discover more stable cognitive capabilities in models. While this approach enhances reliability, its performance benefits remain limited. In practice, it is worthwhile to reconsider uncertainty not only as a hindrance but also as a potential source of exploration and adaptation. This perspective gains support from the observations in Figure 4 (c), where robust models often exhibit noticeable early-stage fluctuations. These fluctuations suggest a possible link between early training uncertainty and latent model potential. Given the computational cost associated with ensemble-based methods, we investigate whether leveraging uncertainty, in conjunction with the performance smoothing metrics, can concurrently improve the performance and reliability of early stopping-based HPO.

**Standard deviation as a measure of uncertainty.** The practice of monitoring performance variation across consecutive epochs is a well-established early stopping strategy in traditional machine learning model training. When the performance change over several consecutive epochs remains below a predefined threshold, the model is considered robust and stable. Building upon this concept, we explore the standard deviation of model performance as a quantifiable measure of uncertainty. By calculating the standard deviation within a narrow window of epochs, we gain insights into the extent of deviation of the performance metric from its mean value within that specific window. A larger standard deviation indicates greater variability or uncertainty in performance across those epochs, while a smaller standard deviation suggests greater stability and consistency in the model's performance during that epoch window. Leveraging standard deviation, we explore a range of metrics:

▶ *Fixed weighting.* The fixed weighting metric combines the means and standard deviations of empirical losses across successive epochs, employing predetermined fixed weights that remain constant throughout the HPO process. Grounded in Proposition 1, for loss metrics governed by a minimization objective, we propose a metric that involves subtracting the standard deviation from the mean, formally expressed as: $\hat{C}_{\mu-\sigma}^t(\gamma) = \hat{\mu}(\gamma, |W|) - \hat{\sigma}(\gamma, |W|)$, where $|W|$ denotes window size, $\hat{\mu}(\gamma, |W|) = \frac{1}{|W|} \sum_{i=1}^{|W|} \hat{f}^{t-i+1}(\gamma)$ represents the mean, and $\hat{\sigma}(\gamma, |W|) = \sqrt{\frac{1}{|W|} \sum_{i=1}^{|W|} (\hat{f}^{t-i+1}(\gamma) - \hat{\mu})^2}$ represents the standard deviation. In this metric, the mean component serves to mitigate the influence of noise, thereby offering a stable measurement of the model's cognitive capability. Conversely, subtracting the standard deviation aims to uncover the lower bound of the loss that the model could achieve, offering insights into model's potential.

▶ *Dynamic weighting.* The dynamic weighting metrics entail the integration of empirical loss means and standard deviations across successive epochs using adjustable weights, as expressed by: $\hat{C}_\theta^t(\gamma) = \hat{\mu}(\gamma, |W|) - \theta \cdot \hat{\sigma}(\gamma, |W|)$, where $\theta$ represents the dynamic weighting parameter. The dynamics of these weights depend on the fluctuations in the model's actual performance. Drawing on the trends reported in the earlier sections on model loss curves and the behavior of uncertainty, we see that uncertainty can initially serve as an indicator of a model's potential in the early stages of training but gradually transform into noise in the later stages. Accordingly, we explore several

dynamic weighting schemes, including linear, logarithmic, and exponential decays, which are formally expressed as: $\theta_{linear} = 1 - \frac{t}{T}$, $\theta_{log} = 1 - \frac{\log(t)}{\log(T)}$, and $\theta_{exp} = e^{-\lambda_{decay}*t}$. Here, $t$ is the current epoch, $T$ is the total epochs for model convergence, and $\lambda_{decay}$ is the decay rate.

Table 2: P-value from a Wilcoxon signed-rank test for the hypothesis that uncertainty integrated metrics outperform *training loss* and *validation loss*. We also give the number of wins and losses of each metric against *training loss* and *validation loss* over 1000 repetitions. $R$ denotes the budget constraint of Hyperband.

| Benchmark | $R$ | | Against *training loss* | | | | | Against *validation loss* | | | | |
|---|---|---|---|---|---|---|---|---|---|---|---|---|
| | | | $C_\mu$ | $C_{\mu-\sigma}$ | $C_{linear}$ | $C_{log}$ | $C_{exp}$ | $C_\mu$ | $C_{\mu-\sigma}$ | $C_{linear}$ | $C_{log}$ | $C_{exp}$ |
| CIFAR-10 | 81 | p-values | 0.54 | 0.15 | 0.03 | 0.06 | 0.36 | $1.1e^{-23}$ | $1.1e^{-20}$ | $3.0e^{-33}$ | $1.6e^{-31}$ | $2.2e^{-25}$ |
| | | w/l | 67/65 | 61/51 | 68/49 | 67/60 | 68/61 | 518/272 | 525/277 | 541/238 | 543/253 | 572/227 |
| | 150 | p-values | $1.7e^{-5}$ | 0.03 | $5.6e^{-4}$ | $2.3e^{-4}$ | $2.0e^{-5}$ | $2.2e^{-15}$ | $5.2e^{-21}$ | $5.8e^{-24}$ | $4.4e^{-19}$ | $9.6e^{-16}$ |
| | | w/l | 181/108 | 128/94 | 166/110 | 171/113 | 180/108 | 425/272 | 457/266 | 457/238 | 437/248 | 429/272 |
| CIFAR-100 | 81 | p-values | 0.04 | 0.16 | 0.008 | 0.07 | 0.06 | $7.6e^{-17}$ | $2.7e^{-20}$ | $9.0e^{-23}$ | $6.4e^{-20}$ | $5.6e^{-19}$ |
| | | w/l | 36/27 | 22/20 | 24/22 | 29/24 | 35/27 | 436/258 | 422/228 | 427/224 | 424/237 | 430/246 |
| | 150 | p-values | 0.13 | 0.27 | 0.06 | 0.05 | 0.07 | $2.0e^{-21}$ | $4.9e^{-15}$ | $1.4e^{-27}$ | $6.7e^{-4}$ | $4.3e^{-21}$ |
| | | w/l | 31/27 | 20/20 | 30/23 | 31/24 | 31/25 | 463/246 | 428/258 | 474/226 | 468/238 | 461/246 |
| ImageNet-16-120 | 81 | p-values | 0.52 | 0.37 | 0.55 | 0.68 | 0.52 | $9.6e^{-27}$ | $8.2e^{-23}$ | $3.4e^{-32}$ | $6.8e^{-30}$ | $6.4e^{-26}$ |
| | | w/l | 17/17 | 15/13 | 15/13 | 17/16 | 26/11 | 356/172 | 370/159 | 376/149 | 382/166 | 351/176 |
| | 150 | p-values | 0.51 | 0.80 | 0.35 | 0.37 | 0.51 | $2.1e^{-17}$ | $1.3e^{-19}$ | $2.9e^{-16}$ | $1.4e^{-17}$ | $6.3e^{-17}$ |
| | | w/l | 11/9 | 5/9 | 7/6 | 8/7 | 11/9 | 308/153 | 323/158 | 304/162 | 312/151 | 307/154 |

Utilizing the Wilcoxon test, we conduct a quantitative evaluation to assess the distinction between metrics incorporating uncertainty as an indicator of model potential and conventional empirical training and validation losses. The detailed results of this evaluation, including the wins and losses observed over 1000 repetitions, are presented in Table 2. More detailed results on various budget constraints are provided in Appendix E.3. Given the rapid convergence of LCBench, the influence of employing uncertainty is limited. We focus our investigations on the tasks in Nas-Bench-201, with a spectrum of budget constraints. The results consistently confirm the benefits of integrating uncertainty as a metric. Our key findings are as follows. (i) Both the fixed and dynamic weighting metrics exhibit noteworthy performance improvements in comparison to validation loss. Notably, the metrics incorporating uncertainty consistently outperform validation loss in nearly 70% of the trials, yielding an average accuracy improvement of 0.38% and maximum improvement of 5.96%. (ii) The dynamic schemes tend to yield higher improvements than the static schemes over the use of validation loss as the metric, demonstrated by the smaller p-values of $C_{linear}$ in most cases. (iii) While the introduction of uncertainty does not yield a substantial improvement in training loss, it does not lead to any performance degradation either. This conclusion is substantiated by the win/loss counts, wherein nearly 98% of the repetitions witness a tie between the metrics incorporating uncertainty and empirical training loss.

***Insight 4: Combining model uncertainty with conventional early stopping metrics is beneficial, especially when the combination adopts dynamic decays to align with the evolving influence of uncertainty in the training process.***

## 5 DISCUSSION AND FUTURE WORK

This paper gives the first known systematic study on early stopping metrics in HPO, and introduces model uncertainty into early stopping in HPO. Our study yields several guidelines for metric selection, benefiting both users and HPO tools: (i) employing training loss for tasks with slow convergence; (ii) utilizing validation loss to address overfitting concerns; (iii) balancing the expressive power of training loss and the generalization power of validation loss by combining both metrics; (iv) introducing model uncertainty into metric design.

Our investigation opens up some future research opportunities. Possible directions include the development of more precise and efficient techniques for characterizing model uncertainty, the creation of more effective approaches for incorporating uncertainty as a metric to guide early stopping decisions, and the integration of uncertainty as a criterion beyond metrics (e.g., becoming part of the inner design of HPO algorithms). Such endeavors hold promise for delivering deeper insights into model selection and budget allocation. Furthermore, this study suggests that it is important to carefully consider the early stopping metrics in future evaluation and design of new early stopping-based HPO algorithms.

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

## A MORE DETAILS ON EXPERIMENTAL SETUP

In addition to the main paper, here we provide supplementary details on the experimental setup.

**HPO algorithms.** To investigate the impact of different metrics under various early stopping configurations, we execute the Hyperband, BOHB, and Sub-sampling (SS) algorithms across a range of budget constraints and filtering ratios.

*Hyperband Li et al. (2017).* An early stopping-based HPO algorithm. It operates by randomly sampling new configurations and allocating more resources to those with promising potential through repeatedly calling successive halving (Jamieson & Talwalkar, 2016), where early stopping takes place.

*BOHB Falkner et al. (2018).* An early stopping-based HPO algorithm that combines Hyperband with Bayesian optimization. BOHB efficiently navigates the hyperparameter space by using both bandit-based resource allocation and Bayesian model-based learning, aiming to find the best hyperparameter configurations for machine learning models.

*Sub-sampling Huang et al. (2022).* An early stopping based HPO algorithm. It evaluates the potential of the configurations based on the sub-samples of observations. Instead of discarding configurations at each early stopping point, It retains model candidates through early stopping stages, enabling potential resumption of training in subsequent phases.

All the three HPO algorithms involve early stopping configurations. As presented in Table 3, the filtering ratios ($\eta$) considered include 2, 3, and 4, corresponding to the selection of 1/2, 1/3, and 1/4 of the models to continue training at each early stopping iteration, respectively. The budget constraint ($R$) denotes the maximum resources allocated to a single configuration, determined based on the maximum number of epochs in the respective benchmark.

Table 3: Parameter settings for Hyperband, BOHB, and SS. $R$ - the maximum amount of resource that can be allocated to a single configuration. $\eta$ - an input that controls the proportion of configurations discarded in each round of Successive Halving (Li et al., 2017).

| | Nas-Bench-201 | LCBench | LogReg | MLP |
|---|---|---|---|---|
| $R$ | 20, 50, 81, 120, 150 | 3, 6, 10, 15, 30 | 37, 111, 333 | 9, 27, 81 |
| $\eta$ | 2, 3, 4 | 2, 3, 4 | 3 | 3 |
| metrics | Training/validation accuracy/loss | | | |

**Benchmarks.** Table 4 compiles information on the datasets, hyperparameters, fidelity, and dataset sizes for Nas-Bench-201, LCBench, LogReg, and MLP.

*LCBench.* A benchmark composed of funnel-shaped MLP networks evaluated across diverse OpenML datasets, as outlined in Table 4. Each model candidate is derived through random sampling of seven hyperparameters. Each model is trained for 50 epochs and the training, validation and testing information is recorded for all epochs. It employs cross-entropy as the loss function for model training.

*Nas-Bench-201.* This benchmark introduces a fixed cell search space illustrated in Figure 7, featuring a DAG composed of only 4 nodes defining the cell architecture. The architecture's search space encompasses approximately 15k distinct configurations. All models are evaluated on the Cifar10, Cifar100, and ImageNet-16-120 datasets, where the number of instances per category is the same in Cifar10 and Cifar100. The training loss function is the cross-entropy.

*LogReg.* This benchmark has two hyperparameters: learning rate and regularization, applied to logistic regression models trained via SGD optimizer. It comprises 625 model configurations evaluated on AutoML datasets.

*MLP.* This benchmark has five hyperparameters: two controlling network depth and width, and three governing batch size, L2 regularization, and Adam's initial learning rate. Following HPOBench Eggensperger et al. (2021), each hyperparameter is discretized into 10 segments. The evaluation involves a grid of 1,000 configurations for each of 30 distinct architectures on the AutoML datasets.

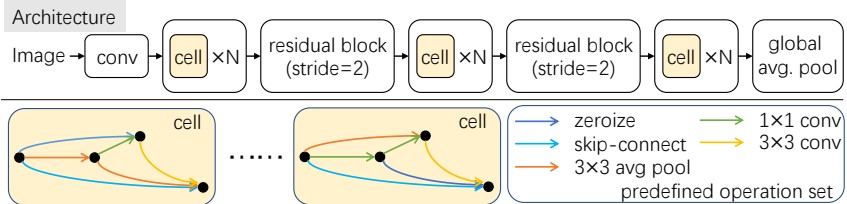

Figure 7: Network architecture in Nas-Bench-201.

Table 4: Details of the benchmarks.

|  | Benchmark | Hyperparameters | Fidelity | # Confs | # Training set | # Validation set | # Test set |
|---|---|---|---|---|---|---|---|
| Nas-Bench-201 | CIFAR-10 (Krizhevsky et al., 2009) | $1 \leftarrow 0$ $2 \leftarrow \{0,1\}^*$ | 1-200 | 15625 | 25K images | 25K images | 10K images |
|  | CIFAR-100 (Krizhevsky et al., 2009) | $3 \leftarrow \{0,1,2\}^*$ Range: {none, skip_connect, nor_conv_1x1, |  |  | 50K images | 5K images | 5K images |
|  | ImageNet-16-120 (Chrabaszcz et al., 2017) | nor_conv_3x3, avg_pool_3x3} |  |  | 151.7K images | 3K images | 3K images |
| LCBench | Fashion-MNIST adult higgs jasmine vehicle volkert (Vanschoren et al., 2014) (Gijsbers et al., 2019) | Batch size: [16, 512], log-scale Learning rate: [$1e^{-4}$, $1e^{-1}$], log-scale Momentum: [0.1, 0.99] Weight decay: [$1e^{-5}$, $1e^{-1}$] Number of layers: [1, 5] Maximum number of units per layer: [64, 1024], log-scale Dropout: [0.0, 1.0] | 1-50 | 2000 | "Whenever possible, we use the given test split with a 33% test split and additionally use fixed 33% of the training data as validation split. In case there is no such OpenML task with a 33% split available for a dataset, we create a 33% test split and fix it across the configurations." (Zimmer et al., 2021) |  |  |
| Tabular | LogReg | alpha: [$1e^{-5}$, 1.0] eta0: [$1e^{-5}$, 1.0] | 1000 | 441 |  |  |  |
|  | MLP | alpha: [$1e^{-8}$, 1.0] batch_size: [4, 256] depth: [1, 3] learning_rate_init: [$1e^{-5}$, 1.0] width: [16, 1024] | 243 | 30k |  |  |  |

**Significance test.** For each benchmark, we run 1000 repetitions employing various metrics within different Hyperband settings. We apply the Wilcoxon signed-rank test, a non-parametric statistical hypothesis test, to assess the significance of differences between the metrics. For significantly

different groups, we quantify the difference between their group means by using standardized effect sizes called Cohen's d (Cohen, 2013), which is calculated by

$$\text{Cohen's d} = \frac{\text{Group A Mean} - \text{Group B Mean}}{\text{Pooled Standard Deviation}}. \qquad (A.1)$$

Additionally, we count the number of wins, ties, and losses of each pair of metrics across the 1000 repetitions.

**Final objective.** We choose test accuracy as the final objective based on several considerations. First, we use balanced accuracy throughout our experiments, except for *ImageNet*. However, our findings demonstrate that using both test loss and test accuracy yields consistent results, especially for ImageNet. Our analysis revealed a substantial linear correlation between final test loss and accuracy, showcased in Figure 8. The disparity in accuracy among models with nearly identical losses (differences smaller than $1e^{-6}$) is approximately 3%. Conversely, for models achieving the same accuracy, the difference in losses remains within 0.4. Specifically, within ImageNet, the observed differences in accuracy and loss are $1.7e^{-7}$% and 0.08, respectively.

Second, the inherent biases and uncertainties within the test dataset render neither test loss nor test accuracy as definitive assessment metrics. While cross-entropy loss provides a nuanced model ranking by capturing subtle differences between predicted and true distributions, it might react variably to minor fluctuations in class probabilities. Although test loss offers depth in assessment, it can also harbor additional noise or uncertainty. Our experiments conducted with test loss as the final objective (Table 5) in comparison to those using test accuracy as the final objective (Table 1) demonstrate an alignment in the significance of differences obtained. Third, test accuracy, due to its simplicity and intuitive nature, aligns more closely with practical application goals. Real-world scenarios such as image classfication competitions and LLM leadearboards predominantly favor test accuracy as the final objective.

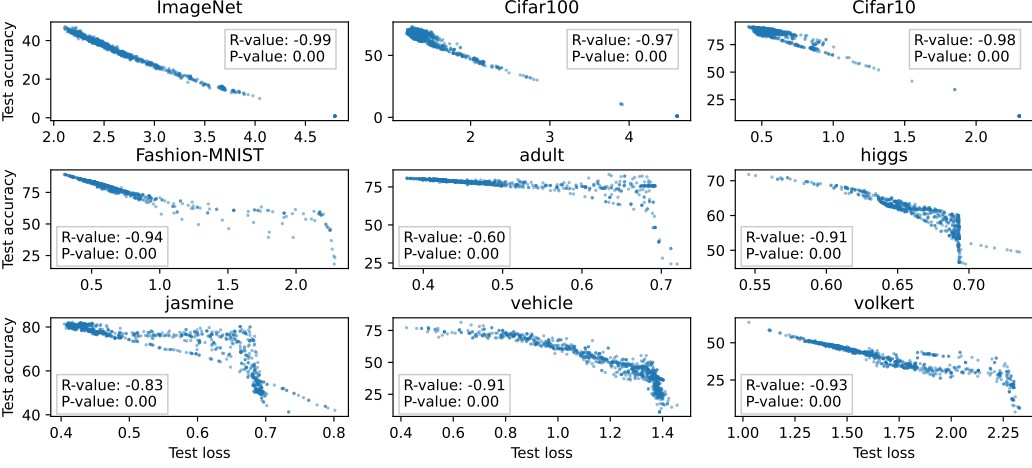

Figure 8: Linear regression between test loss and test accuracy. R-value refers to the Pearson correlation coefficient.

# B MORE RESULTS ON THE RELIABILITY OF COMMON METRICS

## B.1 MORE RESULTS ON HYPERBAND

As an extension of Figure 1, Figure 9 presents the outcomes of additional tasks within LCBench, utilizing common metrics. While not as pronounced as observed in Fashion-MNIST and jasmine, these tasks also indicate a slight superiority of validation metrics over training metrics. This trend aligns with the analysis outlined in Section 3, suggesting a correlation with the rapid convergence of the models.

Table 5: P-value from a Wilcoxon signed-rank test for the hypothesis that *training loss* and *validation loss* outperform commonly used metrics. We also give the number of wins and losses of each metric against *training loss* and *validation loss* over 1000 repetitions. We use **test loss** as the final objective in this set of experiment.

| | CIFAR-10 | | | CIFAR-100 | | | ImageNet-16-120 | | |
|---|---|---|---|---|---|---|---|---|---|
| Against train. loss | Train. acc. | Valid. loss | Valid. acc. | Train. acc. | Valid. loss | Valid. acc. | Train. acc. | Valid. loss | Valid. acc. |
| P-values | 0.08 | $4.9e^{-80}$ | $2.6e^{-40}$ | 0.59 | $1.3e^{-30}$ | $1.4e^{-46}$ | 0.17 | $4.3e^{-35}$ | $6.7e^{-25}$ |
| Wins/losses | 28/35 | 188/601 | 189/683 | 40/30 | 389/281 | 307/251 | 25/27 | 120/332 | 114/279 |

| | Fashion-MNIST | | | jasmine | | | vehicle | | |
|---|---|---|---|---|---|---|---|---|---|
| Against valid. loss | Valid. acc. | Train. loss | Train. acc. | Valid. acc. | Train. loss | Train. acc. | Valid. acc. | Train. loss | Train. acc |
| P-values | $4.7e^{-23}$ | $6.4e^{-17}$ | $3.2e^{-35}$ | $1.7e^{-17}$ | $4.3e^{-66}$ | $1.2e^{-57}$ | $6.0e^{-14}$ | $9.2e^{-5}$ | $2.7e^{-21}$ |
| Wins/losses | 9/124 | 45/159 | 36/244 | 131/335 | 61/463 | 69/444 | 61/165 | 34/80 | 48/178 |

Figure 9: Mean accuracy across 1000 repetitions of optimal configurations selected with commonly used early stopping metrics under diverse budget constraints (upper row). Mean optimal *regret-over-time* filtered with commonly used metrics (lower row).

We present additional results on the comparison of commonly used metrics. First, we report the significance p-values denoting the differences between the metrics in each benchmark under distinct Hyperband settings, along with a tabulation of wins, ties, and losses, presented in Tables 6, 7, 8, 9, 10, 11, 12, 13, and 14. Subsequently, we provide Figures 10 and 11, illustrating the final test accuracy derived from the aggregated results obtained by employing these commonly used metrics across various settings.

Table 6: P-value from a Wilcoxon signed-rank test for commonly used metric pairs on *CIFAR-10*. A p-value below 0.01 indicates that metric $A$ demonstrates superior performance compared to metric $B$ in the "$A$ vs. $B$" comparison. We also give the number of wins and losses where metric $A$ outperforms or lags behind metric $B$. $R$ denotes the budget constraint of Hyperband, and $\eta$ denotes the filtering ratio used by early stopping in Hyperband.

| $\eta$ | | $T_{loss}$ vs. $V_{loss}$ | | | | | $T_{loss}$ vs. $V_{acc}$ | | | | |
|---|---|---|---|---|---|---|---|---|---|---|---|
| | | $R=20$ | $R=50$ | $R=81$ | $R=120$ | $R=150$ | $R=20$ | $R=50$ | $R=81$ | $R=120$ | $R=150$ |
| 2 | p-values | $4.8e^{-88}$ | $1.1e^{-128}$ | $5.0e^{-131}$ | $1.1e^{-131}$ | $9.1e^{-85}$ | $1.1e^{-73}$ | $2.2e^{-105}$ | $1.4e^{-91}$ | $2.4e^{-73}$ | $2.9e^{-12}$ |
| | wins/losses | 588/91 | 762/73 | 798/89 | 796/78 | 689/189 | 498/82 | 644/86 | 644/135 | 582/158 | 425/298 |
| 3 | p-values | $8.7e^{-52}$ | $6.6e^{-109}$ | $7.1e^{-133}$ | $8.8e^{-120}$ | $4.4e^{-85}$ | $5.7e^{-39}$ | $7.4e^{-89}$ | $5.4e^{-106}$ | $5.4e^{-80}$ | $8.4e^{-30}$ |
| | wins/losses | 436/117 | 683/84 | 803/80 | 761/96 | 631/156 | 359/112 | 571/84 | 682/115 | 608/134 | 447/225 |
| 4 | p-values | $1.4e^{-61}$ | $9.7e^{-89}$ | $1.8e^{-129}$ | $9.4e^{-113}$ | $3.9e^{-86}$ | $5.6e^{-41}$ | $7.22e^{-68}$ | $3.6e^{-103}$ | $9.4e^{-113}$ | $3.9e^{-86}$ |
| | wins/losses | 494/116 | 583/93 | 780/52 | 708/86 | 609/140 | 373/110 | 475/91 | 654/87 | 551/103 | 421/179 |

| $\eta$ | | $T_{acc}$ vs. $V_{loss}$ | | | | | $T_{acc}$ vs. $V_{acc}$ | | | | |
|---|---|---|---|---|---|---|---|---|---|---|---|
| | | $R=20$ | $R=50$ | $R=81$ | $R=120$ | $R=150$ | $R=20$ | $R=50$ | $R=81$ | $R=120$ | $R=150$ |
| 2 | p-values | $8.8e^{-90}$ | $4.7e^{-129}$ | $1.6e^{-129}$ | $1.2e^{-129}$ | $1.5e^{-77}$ | $8/3e^{-76}$ | $2.7e^{-105}$ | $1.1e^{-88}$ | $1.3e^{-70}$ | $3.6e^{-7}$ |
| | wins/losses | 593/87 | 769/72 | 792/92 | 781/82 | 661/214 | 501/75 | 648/88 | 632/143 | 569/163 | 395/318 |
| 3 | p-values | $5.5e^{-49}$ | $1.7e^{-110}$ | $6.0e^{-132}$ | $1.3e^{-117}$ | $6.1e^{-82}$ | $1.2e^{-35}$ | $8.4e^{-91}$ | $1.7e^{-104}$ | $4.5e^{-76}$ | $2.3e^{-24}$ |
| | wins/losses | 434/112 | 687/84 | 799/81 | 749/104 | 615/159 | 357/117 | 574/81 | 673/117 | 596/140 | 434/234 |
| 4 | p-values | $4.0e^{-60}$ | $1.8e^{-90}$ | $2.0e^{-128}$ | $1.5e^{-112}$ | $8.8e^{-84}$ | $8.8e^{-39}$ | $7.6e^{-70}$ | $1.7e^{-101}$ | $2.4e^{-76}$ | $1.5e^{-26}$ |
| | wins/losses | 495/118 | 582/87 | 776/164 | 709/88 | 601/137 | 368/113 | 479/84 | 648/98 | 553/112 | 411/190 |

Table 7: P-value from a Wilcoxon signed-rank test for commonly used metric pairs on *CIFAR-100*. A p-value below $0.01$ indicates that metric $A$ demonstrates superior performance compared to metric $B$ in the "$A$ vs. $B$" comparison. We also give the number of wins and losses where metric $A$ outperforms or lags behind metric $B$. $R$ denotes the budget constraint of Hyperband, and $\eta$ denotes the filtering ratio used by early stopping in Hyperband.

| $\eta$ | | $T_{loss}$ vs. $V_{loss}$ | | | | | $T_{loss}$ vs. $V_{acc}$ | | | | |
|---|---|---|---|---|---|---|---|---|---|---|---|
| | | $R=20$ | $R=50$ | $R=81$ | $R=120$ | $R=150$ | $R=20$ | $R=50$ | $R=81$ | $R=120$ | $R=150$ |
| 2 | p-values | $2.8e^{-26}$ | $1.9e^{-89}$ | $2.2e^{-103}$ | $1.3e^{-114}$ | $3.0e^{-134}$ | $3.5e^{-52}$ | $8.5e^{-70}$ | $2.6e^{-73}$ | $9.0e^{-64}$ | $5.6e^{-33}$ |
| | wins/losses | 447/88 | 598/85 | 663/105 | 716/103 | 826/88 | 377/80 | 499/97 | 529/124 | 504/120 | 441/183 |
| 3 | p-values | $1.8e^{-27}$ | $7.0e^{-79}$ | $1.1e^{-98}$ | $1.9e^{-109}$ | $1.0e^{-114}$ | $9.1e^{-29}$ | $1.5e^{-61}$ | $2/9e^{-66}$ | $1.9e^{-64}$ | $7.5e^{-32}$ |
| | wins/losses | 323/130 | 530/94 | 645/92 | 698/97 | 770/120 | 293/102 | 442/93 | 518/115 | 506/131 | 403/209 |
| 4 | p-values | $2/7e^{-41}$ | $1.2e^{-57}$ | $1.1e^{-94}$ | $3.3e^{-104}$ | $8.5e^{-106}$ | $2.4e^{-36}$ | $6.3e^{-44}$ | $3.2e^{-73}$ | $5.8e^{-59}$ | $7.2e^{-28}$ |
| | wins/losses | 371/108 | 438/100 | 614/80 | 664/107 | 705/122 | 301/82 | 342/85 | 501/89 | 468/136 | 391/183 |

| $\eta$ | | $T_{acc}$ vs. $V_{loss}$ | | | | | $T_{acc}$ vs. $V_{acc}$ | | | | |
|---|---|---|---|---|---|---|---|---|---|---|---|
| | | $R=20$ | $R=50$ | $R=81$ | $R=120$ | $R=150$ | $R=20$ | $R=50$ | $R=81$ | $R=120$ | $R=150$ |
| 2 | p-values | $2.2e^{-60}$ | $2.9e^{-88}$ | $3.2e^{-105}$ | $1.4e^{-114}$ | $.5e^{-135}$ | $3.9e^{-50}$ | $7.0e^{-68}$ | $2.6e^{-75}$ | $1.6e^{-65}$ | $5.5e^{-33}$ |
| | wins/losses | 439/87 | 586/91 | 668/96 | 712/102 | 828/88 | 372/80 | 488/103 | 534/114 | 500/112 | 435/186 |
| 3 | p-values | $3.2e^{-26}$ | $3.3e^{-78}$ | $3.3e^{-97}$ | $1.8e^{-109}$ | $2.3e^{-116}$ | $4.4e^{-27}$ | $4.9e^{-60}$ | $1.1e^{-65}$ | $1.6e^{-64}$ | $6.6e^{-24}$ |
| | wins/losses | 318/133 | 523/91 | 634/99 | 696/97 | 768/116 | 286/105 | 438/96 | 505/116 | 507/127 | 393/205 |
| 4 | p-values | $5.6e^{-39}$ | $5.8e^{-55}$ | $3.6e^{-94}$ | $2.6e^{-104}$ | $3.2e^{-105}$ | $2.0e^{-33}$ | $4.6e^{-41}$ | $6.5e^{-73}$ | $4.6e^{-60}$ | $2.0e^{-37}$ |
| | wins/losses | 365/111 | 432/105 | 610/81 | 666/105 | 702/121 | 292/85 | 336/90 | 498/88 | 467/130 | 391/185 |

Table 8: P-value from a Wilcoxon signed-rank test for commonly used metric pairs on *ImageNet-16-120*. A p-value below $0.01$ indicates that metric $A$ demonstrates superior performance compared to metric $B$ in the "$A$ vs. $B$" comparison. We also give the number of wins and losses where metric $A$ outperforms or lags behind metric $B$. $R$ denotes the budget constraint of Hyperband, and $\eta$ denotes the filtering ratio used by early stopping in Hyperband.

| $\eta$ | | $T_{loss}$ vs. $V_{loss}$ | | | | | $T_{loss}$ vs. $V_{acc}$ | | | | |
|---|---|---|---|---|---|---|---|---|---|---|---|
| | | $R=20$ | $R=50$ | $R=81$ | $R=120$ | $R=150$ | $R=20$ | $R=50$ | $R=81$ | $R=120$ | $R=150$ |
| 2 | p-values | $7.3e^{-46}$ | $5.0e^{-66}$ | $1.9e^{-66}$ | $1.7e^{-37}$ | $1.1e^{-39}$ | $7.03^{-35}$ | $9.1e^{-51}$ | $2.9e^{-52}$ | $3.4e^{-33}$ | $8.7e^{-29}$ |
| | wins/losses | 359/84 | 382/93 | 429/113 | 333/119 | 322/140 | 297/86 | 382/93 | 429/113 | 333/119 | 322/140 |
| 3 | p-values | $7.8e^{-41}$ | $2.9e^{-51}$ | $1.4e^{-66}$ | $3.1e^{-32}$ | $7.2e^{-34}$ | $9.6e^{-33}$ | $1.4e^{-41}$ | $1.2e^{-46}$ | $2.6e^{-24}$ | $9.3e^{-27}$ |
| | wins/losses | 320/76 | 388/82 | 493/122 | 384/162 | 382/146 | 259/68 | 332/82 | 403/116 | 329/143 | 338/143 |
| 4 | p-values | $2.7e^{-38}$ | $5.0e^{-47}$ | $1.0e^{-57}$ | $1.3e^{-43}$ | $8.3e^{-26}$ | $4.2e^{-31}$ | $2.7e^{-40}$ | $2.1e^{-40}$ | $2.2e^{-34}$ | $1.1e^{-24}$ |
| | wins/losses | 299/77 | 351/74 | 459/97 | 387/113 | 326/133 | 257/75 | 294/61 | 370/101 | 328/111 | 298/114 |

| $\eta$ | | $T_{acc}$ vs. $V_{loss}$ | | | | | $T_{acc}$ vs. $V_{acc}$ | | | | |
|---|---|---|---|---|---|---|---|---|---|---|---|
| | | $R=20$ | $R=50$ | $R=81$ | $R=120$ | $R=150$ | $R=20$ | $R=50$ | $R=81$ | $R=120$ | $R=150$ |
| 2 | p-values | $1.1e^{-45}$ | $4.7e^{-64}$ | $1.4e^{-63}$ | $1.2e^{-36}$ | $3.1e^{-37}$ | $4.7e^{-34}$ | $9.3e^{-48}$ | $8.2e^{-49}$ | $3.4e^{-32}$ | $1.6e^{-26}$ |
| | wins/losses | 357/90 | 463/89 | 499/109 | 358/122 | 385/134 | 294/93 | 384/103 | 434/119 | 331/119 | 320/144 |
| 3 | p-values | $1.9e^{-37}$ | $8.1e^{-53}$ | $2.7e^{-65}$ | $2.1e^{-30}$ | $8.6e^{-32}$ | $1.6e^{-29}$ | $9.2e^{-43}$ | $1.1e^{-45}$ | $1.6e^{-22}$ | $3.8e^{-25}$ |
| | wins/losses | 308/79 | 396/80 | 491/126 | 382/168 | 384/158 | 246/71 | 338/81 | 402/120 | 329/148 | 335/150 |
| 4 | p-values | $1.4e^{-36}$ | $4.1e^{-46}$ | $1.2e^{-56}$ | $7.1e^{-44}$ | $2.4e^{-23}$ | $1.3e^{-29}$ | $3.3e^{-40}$ | $3.2e^{-39}$ | $3.5e^{-34}$ | $6.4e^{-22}$ |
| | wins/losses | 293/80 | 348/73 | 455/99 | 388/115 | 325/140 | 247/73 | 294/59 | 368/108 | 334/114 | 294/119 |

Table 9: P-value from a Wilcoxon signed-rank test for commonly used metric pairs on *Fashion-MNIST*. A p-value below $0.01$ indicates that metric $A$ demonstrates superior performance compared to metric $B$ in the "$A$ vs. $B$" comparison. We also give the number of wins and losses where metric $A$ outperforms or lags behind metric $B$. $R$ denotes the budget constraint of Hyperband, and $\eta$ denotes the filtering ratio used by early stopping in Hyperband.

| $\eta$ | | $V_{loss}$ vs. $T_{loss}$ | | | | | $V_{loss}$ vs. $T_{acc}$ | | | | |
| --- | --- | --- | --- | --- | --- | --- | --- | --- | --- | --- | --- |
| | | $R=3$ | $R=6$ | $R=10$ | $R=15$ | $R=30$ | $R=3$ | $R=6$ | $R=10$ | $R=15$ | $R=30$ |
| 2 | p-values | $1.7e^{-3}$ | $2.3e^{-12}$ | $3.3e^{-15}$ | $1.1e^{-16}$ | $7.1e^{-8}$ | $9.7e^{-4}$ | $4.7e^{-30}$ | $1.6e^{-35}$ | $2.1e^{-34}$ | $3.9e^{-19}$ |
| | wins/losses | 60/33 | 145/60 | 171/49 | 167/35 | 165/81 | 99/53 | 231/47 | 262/37 | 249/29 | 229/76 |
| 3 | p-values | $1.6e^{-7}$ | $4.8e^{-9}$ | $3.9e^{-24}$ | $2.4e^{-14}$ | $5.0e^{-21}$ | $9.7e^{-11}$ | $1.1e^{-17}$ | $1.0e^{-41}$ | $3.5e^{-32}$ | $6.6e^{-38}$ |
| | wins/losses | 104/42 | 93/30 | 180/39 | 168/48 | 241/76 | 139/48 | 138/27 | 273/36 | 251/39 | 315/69 |
| 4 | p-values | - | $2.0e^{-8}$ | $1.9e^{-8}$ | $4.5e^{-7}$ | $9.3e^{-15}$ | - | $4.7e^{-22}$ | $3.0e^{-18}$ | $5.3e^{-17}$ | $5.1e^{-34}$ |
| | wins/losses | - | 109/51 | 92/34 | 104/36 | 118/6 | - | 181/49 | 143/29 | 166/32 | 279/54 |
| $\eta$ | | $V_{acc}$ vs. $T_{loss}$ | | | | | $V_{acc}$ vs. $T_{acc}$ | | | | |
| | | $R=3$ | $R=6$ | $R=10$ | $R=15$ | $R=30$ | $R=3$ | $R=6$ | $R=10$ | $R=15$ | $R=30$ |
| 2 | p-values | 0.15 | 0.81 | 0.12 | $3.8e^{-3}$ | $9.2e^{-5}$ | $2.8e^{-3}$ | $5.8e^{-11}$ | $1.9e^{-16}$ | $7.8e^{-20}$ | $8.7e^{-16}$ |
| | wins/losses | 53/49 | 83/105 | 104/77 | 111/74 | 135/73 | 69/43 | 151/69 | 179/50 | 176/45 | 195/63 |
| 3 | p-values | 0.04 | 0.41 | $3.7e^{-3}$ | 0.85 | $1.3e^{-5}$ | $3.1e^{-5}$ | $1.1e^{-6}$ | $2.5e^{-22}$ | $3.0e^{-10}$ | $3.1e^{-21}$ |
| | wins/losses | 70/53 | 54/56 | 103/80 | 92.103 | 180/94 | 94/45 | 86/38 | 181/49 | 163/75 | 248/74 |
| 4 | p-values | - | 0.97 | 0.67 | 0.93 | $4.0e^{-3}$ | - | $6.7e^{-7}$ | $1.3e^{-6}$ | $1.6e^{-5}$ | $6.8e^{-20}$ |
| | wins/losses | - | 63/96 | 57/69 | 57/72 | 133/83 | - | 123/76 | 94/46 | 100/49 | 209/54 |

Table 10: P-value from a Wilcoxon signed-rank test for commonly used metric pairs on *adult*. A p-value below $0.01$ indicates that metric $A$ demonstrates superior performance compared to metric $B$ in the "$A$ vs. $B$" comparison. We also give the number of wins and losses where metric $A$ outperforms or lags behind metric $B$. $R$ denotes the budget constraint of Hyperband, and $\eta$ denotes the filtering ratio used by early stopping in Hyperband.

| $\eta$ | | $T_{loss}$ vs. $V_{loss}$ | | | | | $T_{loss}$ vs. $V_{acc}$ | | | | |
| --- | --- | --- | --- | --- | --- | --- | --- | --- | --- | --- | --- |
| | | $R=3$ | $R=6$ | $R=10$ | $R=15$ | $R=30$ | $R=3$ | $R=6$ | $R=10$ | $R=15$ | $R=30$ |
| 2 | p-values | 0.99 | 0.27 | $2.6e^{-16}$ | $1.4e^{-15}$ | $1.3e^{-31}$ | $1.3e^{-3}$ | $3.5e^{-17}$ | $1.1e^{-54}$ | $2.4e^{-22}$ | $2.7e^{-4}$ |
| | wins/losses | 55/87 | 139/129 | 252/103 | 227/97 | 263/87 | 164/140 | 292/149 | 470/121 | 329/141 | 348/227 |
| 3 | p-values | 0.99 | 0.99 | $5.7e^{-6}$ | $1.2e^{-10}$ | $4.6e^{-23}$ | $3.9e^{-3}$ | $5.0e^{-4}$ | $1.4e^{-26}$ | $8.3e^{-27}$ | $1.6e^{-11}$ |
| | wins/losses | 66/115 | 54/94 | 226/147 | 201/104 | 288/112 | 186/156 | 198/145 | 382/160 | 350/135 | 387/205 |
| 4 | p-values | - | 0.98 | 0.99 | 0.28 | $2.7e^{-7}$ | - | $1.1e^{-8}$ | $4.2e^{-6}$ | 0.03 | $3.5e^{-7}$ |
| | wins/losses | - | 86/105 | 88/109 | 88/81 | 193/125 | - | 232/136 | 218/142 | 178/150 | 290/189 |
| $\eta$ | | $T_{acc}$ vs. $V_{loss}$ | | | | | $T_{acc}$ vs. $V_{acc}$ | | | | |
| | | $R=3$ | $R=6$ | $R=10$ | $R=15$ | $R=30$ | $R=3$ | $R=6$ | $R=10$ | $R=15$ | $R=30$ |
| 2 | p-values | 0.99 | 0.98 | $1.1e^{-3}$ | $1.1e^{-8}$ | $6.5e^{-36}$ | 0.027 | $1.4e^{-9}$ | $9.7e^{-40}$ | $2.6e^{-20}$ | $1.2e^{-8}$ |
| | wins/losses | 94/140 | 178/210 | 275/208 | 293/172 | 340/114 | 154/131 | 267/173 | 421/145 | 322/138 | 347/168 |
| 3 | p-values | 0.99 | 0.99 | $2.5e^{-3}$ | $7.0e^{-5}$ | $1.3e^{-21}$ | 0.61 | $9.5e^{-3}$ | $1.6e^{-22}$ | $3.7e^{-22}$ | $7.7e^{-15}$ |
| | wins/losses | 111/178 | 109/161 | 264/200 | 263/187 | 345/164 | 170/164 | 178/138 | 370/162 | 336/151 | 369/175 |
| 4 | p-values | - | 0.99 | 0.99 | 0.47 | $9.2e^{-11}$ | - | $5.7e^{-7}$ | $4.3e^{-3}$ | 0.07 | $1.3e^{-11}$ |
| | wins/losses | - | 116/153 | 312/174 | 142/135 | 265/164 | - | 218/126 | 191/138 | 162/138 | 285/151 |

Table 11: P-value from a Wilcoxon signed-rank test for commonly used metric pairs on *higgs*. A p-value below $0.01$ indicates that metric $A$ demonstrates superior performance compared to metric $B$ in the "$A$ vs. $B$" comparison. We also give the number of wins and losses where metric $A$ outperforms or lags behind metric $B$. $R$ denotes the budget constraint of Hyperband, and $\eta$ denotes the filtering ratio used by early stopping in Hyperband.

| $\eta$ | | $V_{loss}$ vs. $T_{loss}$ | | | | | $V_{loss}$ vs. $T_{acc}$ | | | | |
|---|---|---|---|---|---|---|---|---|---|---|---|
| | | $R=3$ | $R=6$ | $R=10$ | $R=15$ | $R=30$ | $R=3$ | $R=6$ | $R=10$ | $R=15$ | $R=30$ |
| 2 | p-values | $4.2e^{-11}$ | $3.1e^{-14}$ | $3.1e^{-10}$ | $1.6e^{-13}$ | $5.4e^{-6}$ | $6.5e^{-8}$ | $1.3e^{-5}$ | $2.5e^{-7}$ | $1.4e^{-3}$ | $9.7e^{-3}$ |
| | wins/losses | 73/13 | 82/18 | 105/36 | 98/23 | 83/39 | 108/42 | 123/74 | 146/91 | 131/95 | 112/77 |
| 3 | p-values | $9.6e^{-14}$ | $5.3e^{-13}$ | $3.4e^{-9}$ | $2.6e^{-12}$ | 0.01 | $1.2e^{-13}$ | $5.0e^{-10}$ | $2.7e^{-4}$ | $7.7e^{-3}$ | $1.5e^{-3}$ |
| | wins/losses | 120/34 | 75/13 | 106/28 | 82/28 | 75/39 | 134/43 | 107/43 | 151/107 | 115/108 | 123/78 |
| 4 | p-values | - | $1.1e^{-13}$ | $2.0e^{-11}$ | $1.5e^{-6}$ | $1.9e^{-13}$ | - | $4.2e^{-14}$ | $1.8e^{-9}$ | 0.01 | $5.9e^{-4}$ |
| | wins/losses | - | 88/19 | 81/18 | 45/12 | 95/23 | - | 147/58 | 124/48 | 93/43 | 151/24 |

| $\eta$ | | $V_{acc}$ vs. $T_{loss}$ | | | | | $V_{acc}$ vs. $T_{acc}$ | | | | |
|---|---|---|---|---|---|---|---|---|---|---|---|
| | | $R=3$ | $R=6$ | $R=10$ | $R=15$ | $R=30$ | $R=3$ | $R=6$ | $R=10$ | $R=15$ | $R=30$ |
| 2 | p-values | $2.2e^{-4}$ | $5.0e^{-26}$ | $2.0e^{-18}$ | $1.3e^{-22}$ | $1.1e^{-11}$ | $1.2e^{-4}$ | $1.3e^{-22}$ | $1.4e^{-17}$ | $3.6e^{-16}$ | $5.0e^{-11}$ |
| | wins/losses | 113/80 | 219/75 | 189/71 | 187/50 | 121/45 | 98/53 | 166/44 | 147/42 | 128/32 | 108/34 |
| 3 | p-values | $3.5e^{-5}$ | $4.8e^{-9}$ | $1.1e^{-19}$ | $1.2e^{-25}$ | $5.4e^{-8}$ | $8.5e^{-7}$ | $3.6e^{-11}$ | $6.2e^{-15}$ | $1.2e^{-17}$ | $1.3e^{-11}$ |
| | wins/losses | 119/103 | 131/71 | 220/77 | 195/62 | 122/52 | 105/80 | 108/48 | 159/53 | 121/42 | 137/45 |
| 4 | p-values | - | $1.1e^{-8}$ | $3.5e^{-11}$ | $9.4e^{-10}$ | $2.6e^{-26}$ | - | $1.6e^{-14}$ | $2.4e^{-13}$ | $1.4e^{-9}$ | $4.4e^{-20}$ |
| | wins/losses | - | 153/93 | 151/77 | 136/81 | 200/33 | - | 147/58 | 124/48 | 93/43 | 151/24 |

Table 12: P-value from a Wilcoxon signed-rank test for commonly used metric pairs on *jasmine*. A p-value below $0.01$ indicates that metric $A$ demonstrates superior performance compared to metric $B$ in the "$A$ vs. $B$" comparison. We also give the number of wins and losses where metric $A$ outperforms or lags behind metric $B$. $R$ denotes the budget constraint of Hyperband, and $\eta$ denotes the filtering ratio used by early stopping in Hyperband.

| $\eta$ | | $V_{loss}$ vs. $T_{loss}$ | | | | | $V_{loss}$ vs. $T_{acc}$ | | | | |
|---|---|---|---|---|---|---|---|---|---|---|---|
| | | $R=3$ | $R=6$ | $R=10$ | $R=15$ | $R=30$ | $R=3$ | $R=6$ | $R=10$ | $R=15$ | $R=30$ |
| 2 | p-values | $1.5e^{-4}$ | 0.99 | $9.7e^{-5}$ | $1.2e^{-17}$ | $1.5e^{-92}$ | 0.44 | 0.99 | $2.9e^{-3}$ | $3.6e^{-18}$ | $1.6e^{-87}$ |
| | wins/losses | 71/35 | 94/135 | 286/244 | 366/251 | 692/155 | 70/69 | 108/145 | 276/242 | 365/230 | 670/161 |
| 3 | p-values | $1.9e^{-3}$ | 0.065 | 0.42 | 0.028 | $3.1e^{-82}$ | 0.78 | 0.99 | 0.50 | $8.0e^{-3}$ | $2.1e^{-72}$ |
| | wins/losses | 100/67 | 74/57 | 188/211 | 247/268 | 643/164 | 88/100 | 78/96 | 189/211 | 250/246 | 604/176 |
| 4 | p-values | - | 0.94 | 0.13 | 0.75 | $5.6e^{-30}$ | - | 0.96 | 0.83 | 0.85 | $1.0e^{-34}$ |
| | wins/losses | - | 65/70 | 98/92 | 101/140 | 448/229 | - | 86/98 | 107/127 | 112/149 | 463/225 |

| $\eta$ | | $V_{acc}$ vs. $T_{loss}$ | | | | | $V_{acc}$ vs. $T_{acc}$ | | | | |
|---|---|---|---|---|---|---|---|---|---|---|---|
| | | $R=3$ | $R=6$ | $R=10$ | $R=15$ | $R=30$ | $R=3$ | $R=6$ | $R=10$ | $R=15$ | $R=30$ |
| 2 | p-values | 0.031 | 0.14 | $2.9e^{-3}$ | $3.6e^{-18}$ | $1.6e^{-87}$ | 0.90 | 0.37 | $1.7e^{-46}$ | $3.0e^{-53}$ | $9.4e^{-139}$ |
| | wins/losses | 125/94 | 165/142 | 427/140 | 466/165 | 836/46 | 84/93 | 155/127 | 426/125 | 473/155 | 827/54 |
| 3 | p-values | 0.046 | $2.6e^{-4}$ | $1.7e^{-17}$ | $1.7e^{-21}$ | $1.7e^{-129}$ | 0.98 | 0.18 | $1.4e^{-17}$ | $4.6e^{-23}$ | $2.7e^{124}$ |
| | wins/losses | 131/118 | 170/106 | 292/163 | 337/188 | 793/75 | 81/121 | 134/105 | 288/144 | 333/168 | 767/93 |
| 4 | p-values | - | 0.73 | $8.3e^{-4}$ | $3.8e^{-7}$ | $3.7e^{-89}$ | - | 0.95 | 0.047 | $8.7e^{-7}$ | $5.9e^{95}$ |
| | wins/losses | - | 118/138 | 171/126 | 166/129 | 620/99 | - | 114/141 | 140/118 | 150/110 | 623/92 |

Table 13: P-value from a Wilcoxon signed-rank test for commonly used metric pairs on *vehicle*. A p-value below $0.01$ indicates that metric $A$ demonstrates superior performance compared to metric $B$ in the "$A$ vs. $B$" comparison. We also give the number of wins and losses where metric $A$ outperforms or lags behind metric $B$. $R$ denotes the budget constraint of Hyperband, and $\eta$ denotes the filtering ratio used by early stopping in Hyperband.

| $\eta$ | | $V_{loss}$ vs. $T_{loss}$ | | | | | $V_{loss}$ vs. $T_{acc}$ | | | | |
|---|---|---|---|---|---|---|---|---|---|---|---|
| | | $R=3$ | $R=6$ | $R=10$ | $R=15$ | $R=30$ | $R=3$ | $R=6$ | $R=10$ | $R=15$ | $R=30$ |
| 2 | p-values | $3.0e^{-8}$ | $4.7e^{-10}$ | $6.3e^{-4}$ | $0.053$ | $0.54$ | $6.1e^{-10}$ | $1.3e^{-19}$ | $3.3e^{-11}$ | $3.3e^{-7}$ | $0.004$ |
| | wins/losses | 114/62 | 80/23 | 61/41 | 56/40 | 53/55 | 178/89 | 152/39 | 154/76 | 150/85 | 191/151 |
| 3 | p-values | $1.4e^{-15}$ | $8.5e^{-9}$ | $1.1e^{-6}$ | $0.44$ | $8.6e^{-3}$ | $3.0e^{-14}$ | $4.2e^{-16}$ | $8.2e^{-19}$ | $2.1e^{-6}$ | $1.7e^{-9}$ |
| | wins/losses | 177/59 | 99/41 | 82/41 | 45/48 | 77/54 | 235/102 | 172/54 | 178/60 | 133/83 | 231/133 |
| 4 | p-values | - | $8.5e^{-9}$ | $1.1e^{-10}$ | $4.6e^{-3}$ | $0.048$ | - | $6.4e^{-14}$ | $1.8e^{-18}$ | $.8e^{-9}$ | $5.2e^{-6}$ |
| | wins/losses | - | 88/34 | 82/28 | 44/22 | 53/40 | - | 174/66 | 160/44 | 114/45 | 162/101 |

| $\eta$ | | $V_{acc}$ vs. $T_{loss}$ | | | | | $V_{acc}$ vs. $T_{acc}$ | | | | |
|---|---|---|---|---|---|---|---|---|---|---|---|
| | | $R=3$ | $R=6$ | $R=10$ | $R=15$ | $R=30$ | $R=3$ | $R=6$ | $R=10$ | $R=15$ | $R=30$ |
| 2 | p-values | $2.2e^{-6}$ | $0.063$ | $0.99$ | $0.99$ | $6.4e^{-25}$ | $2.5e^{-12}$ | $1.6e^{-10}$ | $0.069$ | $0.99$ | $4.1e^{-33}$ |
| | wins/losses | 127/78 | 98/90 | 98/150 | 123/171 | 328/136 | 125/44 | 139/68 | 127/141 | 157/154 | 349/117 |
| 3 | p-values | $1.1e^{-12}$ | $0.019$ | $0.32$ | $0.99$ | $1.7e^{-9}$ | $2.4e^{-22}$ | $5.1e^{-10}$ | $3.5e^{-9}$ | $0.1$ | $4.7e^{-24}$ |
| | wins/losses | 190/92 | 100/89 | 110/115 | 91/138 | 274/143 | 170/49 | 120/57 | 170/100 | 138/129 | 331/128 |
| 4 | p-values | - | $6.9e^{-3}$ | $9.9e^{-4}$ | $0.38$ | $2.1e^{-8}$ | - | $8.6e^{-11}$ | $6.1e^{-15}$ | $1.3e^{-7}$ | $8.3e^{-16}$ |
| | wins/losses | - | 109/83 | 96/67 | 72/81 | 200/103 | - | 138/62 | 130/46 | 98/54 | 251/104 |

Table 14: P-value from a Wilcoxon signed-rank test for commonly used metric pairs on *volkert*. A p-value below $0.01$ indicates that metric $A$ demonstrates superior performance compared to metric $B$ in the "$A$ vs. $B$" comparison. We also give the number of wins and losses where metric $A$ outperforms or lags behind metric $B$. $R$ denotes the budget constraint of Hyperband, and $\eta$ denotes the filtering ratio used by early stopping in Hyperband.

| $\eta$ | | $V_{loss}$ vs. $T_{loss}$ | | | | | $V_{loss}$ vs. $T_{acc}$ | | | | |
|---|---|---|---|---|---|---|---|---|---|---|---|
| | | $R=3$ | $R=6$ | $R=10$ | $R=15$ | $R=30$ | $R=3$ | $R=6$ | $R=10$ | $R=15$ | $R=30$ |
| 2 | p-values | $2.0e^{-4}$ | $1.8e^{-9}$ | $1.5e^{-6}$ | $7.9e^{-8}$ | $9.1e^{-4}$ | $3.7e^{-4}$ | $3.7e^{-8}$ | $1.7e^{-7}$ | $5.9e^{-11}$ | $3.5e^{-3}$ |
| | wins/losses | 50/23 | 120/47 | 151/86 | 160/92 | 150/81 | 73/46 | 148/79 | 195/115 | 185/102 | 152/96 |
| 3 | p-values | $1.1e^{-5}$ | $8.8e^{-5}$ | $1.8e^{-11}$ | $2.0e^{-8}$ | $7.0e^{-9}$ | $1.2e^{-4}$ | $1.3e^{-3}$ | $1.7e^{-19}$ | $6.9e^{-9}$ | $6.3e^{-11}$ |
| | wins/losses | 65/31 | 62/28 | 157/71 | 137/76 | 171/72 | 86/43 | 71/48 | 181/105 | 167/105 | 192/86 |
| 4 | p-values | - | $2.6e^{-5}$ | $2.3e^{-7}$ | $2.3e^{-3}$ | $3.8e^{-5}$ | - | $5.5e^{-7}$ | $1.1e^{-5}$ | $6.7e^{-3}$ | $8.7e^{-6}$ |
| | wins/losses | - | 81/41 | 75/29 | 63/38 | 150/78 | - | 102/55 | 96/54 | 84/58 | 184/115 |

| $\eta$ | | $V_{acc}$ vs. $T_{loss}$ | | | | | $V_{acc}$ vs. $T_{acc}$ | | | | |
|---|---|---|---|---|---|---|---|---|---|---|---|
| | | $R=3$ | $R=6$ | $R=10$ | $R=15$ | $R=30$ | $R=3$ | $R=6$ | $R=10$ | $R=15$ | $R=30$ |
| 2 | p-values | $0.42$ | $0.21$ | $0.39$ | $0.63$ | $0.04$ | $0.13$ | $0.17$ | $0.14$ | $0.22$ | $0.12$ |
| | wins/losses | 91/79 | 144/132 | 165/165 | 165/166 | 135/92 | 88/73 | 127/119 | 165/150 | 154/157 | 122/96 |
| 3 | p-values | $0.51$ | $0.99$ | $0.014$ | $0.99$ | $0.064$ | $0.47$ | $0.99$ | $8.4e^{-3}$ | $0.96$ | $0.99$ |
| | wins/losses | 82/90 | 75/100 | 198/161 | 125/165 | 168/108 | 83/77 | 65/99 | 175/142 | 128/169 | 163/101 |
| 4 | p-values | - | $0.99$ | $0.69$ | $0.93$ | $0.78$ | - | $0.95$ | $0.73$ | $0.91$ | $0.56$ |
| | wins/losses | - | 91/127 | 104/99 | 91/110 | 149/142 | - | 79/106 | 94/94 | 89/104 | 146/145 |

Table 15: P-value from a Wilcoxon signed-rank test for commonly used metric pairs on **_LogReg_**. A p-value below $0.01$ indicates that metric $A$ demonstrates superior performance compared to metric $B$ in the "$A$ vs. $B$" comparison. We also give the number of wins and losses where metric $A$ outperforms or lags behind metric $B$. $R$ denotes the budget constraint of Hyperband, and $\eta$ denotes the filtering ratio used by early stopping in Hyperband.

| $\eta = 3$ | $T_{loss}$ vs. $V_{loss}$ | | | $T_{loss}$ vs. $V_{acc}$ | | |
|---|---|---|---|---|---|---|
| | $R = 37$ | $R = 111$ | $R = 333$ | $R = 37$ | $R = 111$ | $R = 333$ |
| p-values | $1.7e^{-3}$ | $7.2e^{-39}$ | $0.99$ | $2.0e^{-3}$ | $1.1e^{-39}$ | $0.99$ |
| wins/losses | $59/57$ | $424/113$ | $6/33$ | $59/58$ | $434/114$ | $6/31$ |

| $\eta = 3$ | $T_{acc}$ vs. $V_{loss}$ | | | $T_{acc}$ vs. $V_{acc}$ | | |
|---|---|---|---|---|---|---|
| | $R = 37$ | $R = 111$ | $R = 333$ | $R = 37$ | $R = 111$ | $R = 333$ |
| p-values | $2.0e^{-3}$ | $1.1e^{-39}$ | $0.99$ | $1.7e^{-3}$ | $7.2e^{-39}$ | $0.99$ |
| wins/losses | $59/57$ | $424/113$ | $6/32$ | $59/58$ | $434/114$ | $6/30$ |

Table 16: P-value from a Wilcoxon signed-rank test for commonly used metric pairs on **_MLP_**. A p-value below $0.01$ indicates that metric $A$ demonstrates superior performance compared to metric $B$ in the "$A$ vs. $B$" comparison. We also give the number of wins and losses where metric $A$ outperforms or lags behind metric $B$. $R$ denotes the budget constraint of Hyperband, and $\eta$ denotes the filtering ratio used by early stopping in Hyperband.

| $\eta = 3$ | $T_{loss}$ vs. $V_{loss}$ | | | $T_{loss}$ vs. $V_{acc}$ | | |
|---|---|---|---|---|---|---|
| | $R = 9$ | $R = 27$ | $R = 81$ | $R = 9$ | $R = 27$ | $R = 81$ |
| p-values | $5.6e^{-59}$ | $6.6e^{-54}$ | $2.6e^{-40}$ | $5.7e^{-59}$ | $1.5e^{-54}$ | $8.3e^{-41}$ |
| wins/losses | $654/238$ | $675/252$ | $632/299$ | $655/239$ | $677/249$ | $634/296$ |

| $\eta = 3$ | $T_{acc}$ vs. $V_{loss}$ | | | $T_{acc}$ vs. $V_{acc}$ | | |
|---|---|---|---|---|---|---|
| | $R = 9$ | $R = 27$ | $R = 81$ | $R = 9$ | $R = 27$ | $R = 81$ |
| p-values | $6.7e^{-63}$ | $1.0e^{-56}$ | $8.8e^{-38}$ | $7.1e^{-63}$ | $3.4e^{-57}$ | $6.1e^{-38}$ |
| wins/losses | $650/226$ | $672/250$ | $620/301$ | $648/225$ | $674/249$ | $619/301$ |

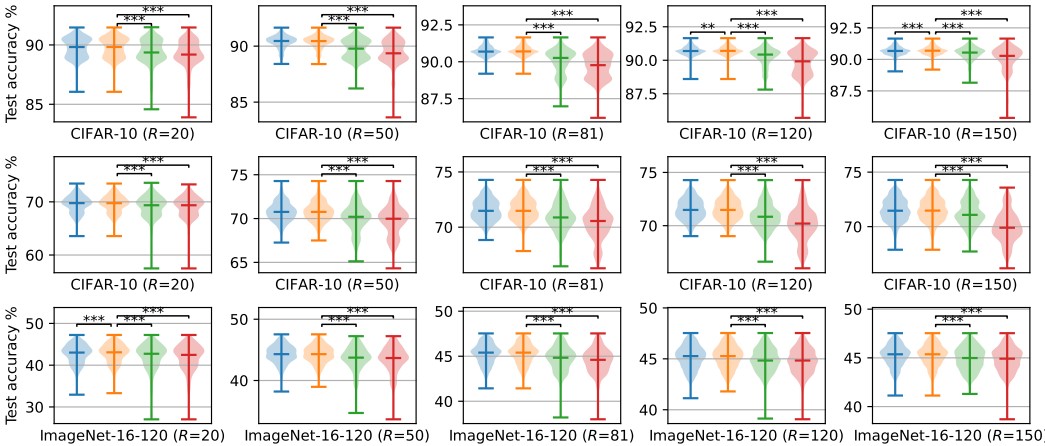

Figure 10: Final test accuracy achieved using different metrics with Hyperband ($\eta = 3$) across diverse budgets ($R$) on Nas-Bench-201. *=p-value<0.05, **=p-value<0.01, ***=p-value<0.001.

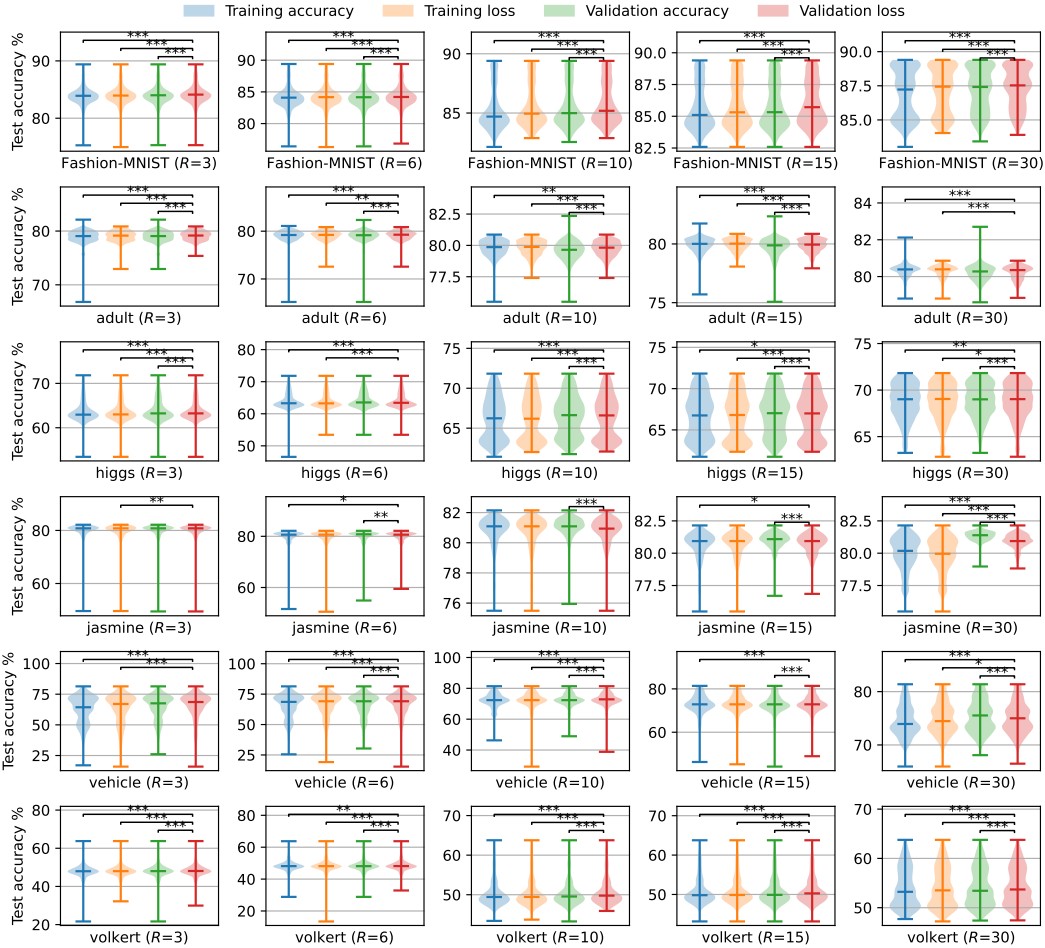

Figure 11: Final test accuracy achieved using different metrics with Hyperband ($\eta = 3$) across diverse budgets ($R$) on LCBench. *=p-value<0.05, **=p-value<0.01, ***=p-value<0.001.

### B.2 RESULTS ON BOHB

We illustrate the average test accuracies and regret values achieved under different budget constraints using the BOHB algorithm in Figures 12 and 13. The top row demonstrates the average test accuracies, while the bottom row showcases the regret values of remaining model configurations after employing various metrics during the HPO process. The experimental configurations are identical to Hyperband. The results align closely with the insights presented in Section 2.2. In Nas-Bench-201, marked by high task complexity, the training metrics consistently outperform the validation metrics in test accuracy, exhibiting an average difference of 0.70% and a maximum difference of 18.37%. Conversely, in LCBench with lower complexity, the validation metrics tend to outperform the training metrics in most cases, with a mean difference of 0.28% and a maximum difference of 15.53%.

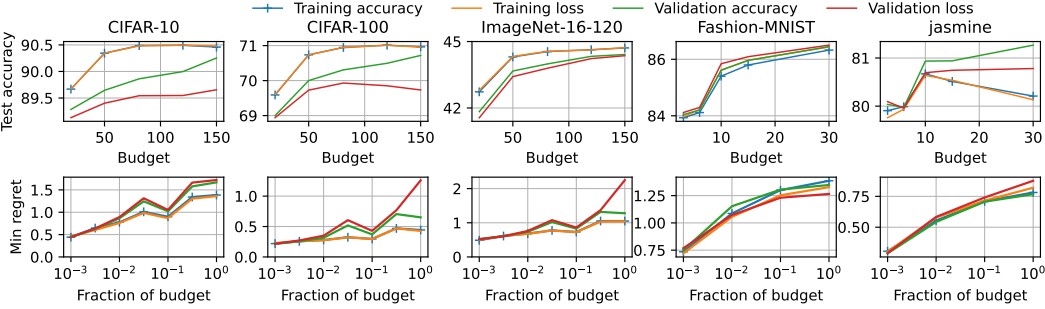

Figure 12: Mean accuracy across 1000 repetitions of optimal configurations selected with commonly used early stopping metrics under diverse budget constraints (upper row). Mean optimal *regret-over-time* filtered with commonly used metrics (lower row). "Fraction of budget" denotes the proportion of allocated budget used during training. This set of experiments is conducted on the BOHB algorithm.

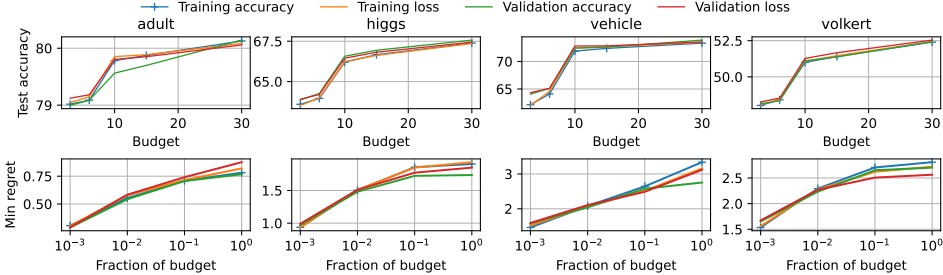

Figure 13: Mean accuracy across 1000 repetitions of optimal configurations selected with commonly used early stopping metrics under diverse budget constraints (upper row). Mean optimal *regret-over-time* filtered with commonly used metrics (lower row). "Fraction of budget" denotes the proportion of allocated budget used during training. This set of experiments is conducted on the BOHB algorithm.

Table 17: P-value from a Wilcoxon signed-rank test for commonly used metric pairs on **LogReg**. A p-value below $0.01$ indicates that metric $A$ demonstrates superior performance compared to metric $B$ in the "$A$ vs. $B$" comparison. We also give the number of wins and losses where metric $A$ outperforms or lags behind metric $B$. $R$ denotes the budget constraint of **BOHB**, and $\eta$ denotes the filtering ratio used by early stopping in BOHB.

| $\eta = 3$ | $T_{loss}$ vs. $V_{loss}$ | | | $T_{loss}$ vs. $V_{acc}$ | | |
|---|---|---|---|---|---|---|
| | $R = 37$ | $R = 111$ | $R = 333$ | $R = 37$ | $R = 111$ | $R = 333$ |
| p-values | $1.1e^{-4}$ | $3.2e^{-4}$ | $1.0$ | $1.3e^{-5}$ | $2.5e^{-5}$ | $1.0$ |
| wins/losses | 85/55 | 109/86 | 79/208 | 93/57 | 114/82 | 74/198 |
| $\eta = 3$ | $T_{acc}$ vs. $V_{loss}$ | | | $T_{acc}$ vs. $V_{acc}$ | | |
| | $R = 37$ | $R = 111$ | $R = 333$ | $R = 37$ | $R = 111$ | $R = 333$ |
| p-values | $2.9e^{-5}$ | $1.0e^{-4}$ | $1.0$ | $2.4e^{-6}$ | $2.3e^{-6}$ | $1.0$ |
| wins/losses | 85/55 | 109/86 | 79/208 | 98/53 | 118/80 | 73/196 |

Table 18: P-value from a Wilcoxon signed-rank test for commonly used metric pairs on **MLP**. A p-value below $0.01$ indicates that metric $A$ demonstrates superior performance compared to metric $B$ in the "$A$ vs. $B$" comparison. We also give the number of wins and losses where metric $A$ outperforms or lags behind metric $B$. $R$ denotes the budget constraint of **BOHB**, and $\eta$ denotes the filtering ratio used by early stopping in BOHB.

| $\eta = 3$ | $T_{loss}$ vs. $V_{loss}$ | | | $T_{loss}$ vs. $V_{acc}$ | | |
|---|---|---|---|---|---|---|
| | $R = 9$ | $R = 27$ | $R = 81$ | $R = 9$ | $R = 27$ | $R = 81$ |
| p-values | $9.5e^{-63}$ | $1.8e^{-54}$ | $1.8e^{-48}$ | $1.8e^{-62}$ | $2.1e^{-54}$ | $3.6e^{-49}$ |
| wins/losses | 657/220 | 665/265 | 662/281 | 654/222 | 665/263 | 672/273 |
| $\eta = 3$ | $T_{acc}$ vs. $V_{loss}$ | | | $T_{acc}$ vs. $V_{acc}$ | | |
| | $R = 9$ | $R = 27$ | $R = 81$ | $R = 9$ | $R = 27$ | $R = 81$ |
| p-values | $1.0e^{-66}$ | $4.0e^{-49}$ | $2.4e^{-53}$ | $1.7e^{-66}$ | $1.8e^{-48}$ | $1.1e^{-53}$ |
| wins/losses | 656/220 | 647/287 | 697/255 | 655/219 | 647/286 | 688/262 |

### B.3 RESULTS ON SS

We illustrate the average test accuracies and regret values achieved under different budget constraints using the SS algorithm in Figures 14 and 15. The top row demonstrates the average test accuracies, while the bottom row showcases the regret values of remaining model configurations after employing various metrics during the HPO process. The experimental configurations are identical to Hyperband. The results align closely with the insights presented in Section 2.2. In Nas-Bench-201, marked by high task complexity, the training metrics consistently outperform the validation metrics in test accuracy, exhibiting an average difference of 0.73% and a maximum difference of 15.47%. Conversely, in LCBench with lower complexity, the validation metrics tend to outperform the training metrics in most cases, with a mean difference of 0.17% and a maximum difference of 5.02%.

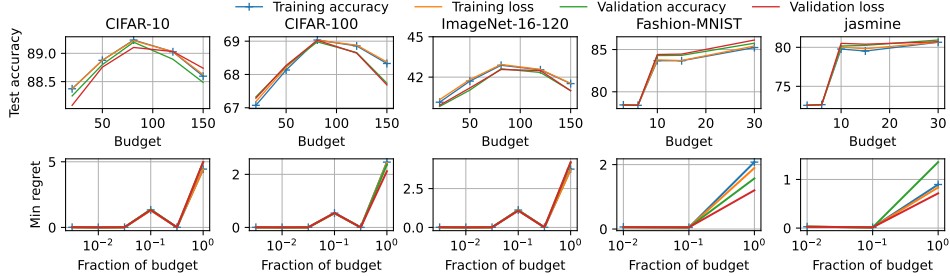

Figure 14: Mean accuracy across 1000 repetitions of optimal configurations selected with commonly used early stopping metrics under diverse budget constraints (upper row). Mean optimal *regret-over-time* filtered with commonly used metrics (lower row). "Fraction of budget" denotes the proportion of allocated budget used during training. This set of experiments is conducted on the SS algorithm.

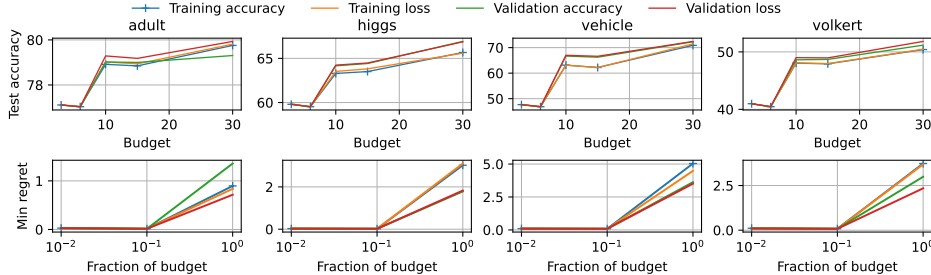

Figure 15: Mean accuracy across 1000 repetitions of optimal configurations selected with commonly used early stopping metrics under diverse budget constraints (upper row). Mean optimal *regret-over-time* filtered with commonly used metrics (lower row). "Fraction of budget" denotes the proportion of allocated budget used during training. This set of experiments is conducted on the SS algorithm.

## C   PROOF OF PROPOSITION 1

*Proof.* First, by utilizing Markov's inequality, we can establish a general upper bound:

$$P(\hat{f}^t(\gamma_o) - \mathbb{E}\hat{f}^t(\gamma_{so}) \geq 0) = P(\hat{f}^t(\gamma_o) \geq \mathbb{E}\hat{f}^t(\gamma_{so})) \leq \frac{\mathbb{E}\hat{f}^t(\gamma_o)}{\mathbb{E}\hat{f}^t(\gamma_{so})} = \frac{f^t(\gamma_o)}{f^t(\gamma_{so})}.$$

To formulate a tighter bound, we introduce the expected risk $f^t$,

$$\begin{aligned} &P(\hat{f}^t(\gamma_o) - \mathbb{E}\hat{f}^t(\gamma_{so}) \geq 0) \\ &= P(\hat{f}^t(\gamma_o) - f^t(\gamma_o) + f^t(\gamma_o) - \mathbb{E}\hat{f}^t(\gamma_{so}) \geq 0) \\ &= P(\hat{f}^t(\gamma_o) - f^t(\gamma_o) \geq f^t(\gamma_{so}) - f^t(\gamma_o)). \end{aligned}$$

Based on Hoeffding's inequality (Hoeffding, 1994), the probability that the average of bounded independent variables $\hat{f}^t(\gamma_o)$ deviates its expected value $f^t(\gamma_o)$ satisfy

$$P(\hat{f}^t(\gamma_o) - f^t(\gamma_o) \geq \epsilon) \leq e^{-\frac{2|D|\epsilon^2}{(ub-lb)^2}}, \quad \forall \epsilon > 0.$$

For the case of $f^t(\gamma_{so}) - f^t(\gamma_o) > 0$, let $\epsilon = f^t(\gamma_{so}) - f^t(\gamma_o)$, the probability becomes

$$P(\hat{f}^t(\gamma_o) - f^t(\gamma_o) \geq f^t(\gamma_{so}) - f^t(\gamma_o)) \leq e^{-\frac{2|D|\left(f^t(\gamma_{so}) - f^t(\gamma_o)\right)^2}{(ub-lb)^2}}.$$

For the case of $f^t(\gamma_{so}) - f^t(\gamma_o) \leq 0$, we have

$$\begin{aligned} &P(\hat{f}^t(\gamma_o) - f^t(\gamma_o) \geq f^t(\gamma_{so}) - f^t(\gamma_o)) \\ &= 1 - P(\hat{f}^t(\gamma_o) - f^t(\gamma_o) \leq f^t(\gamma_{so}) - f^t(\gamma_o)). \end{aligned}$$

Similarly, based on Hoeffding's inequality, the probability satisfy

$$P(\hat{f}^t(\gamma_o) - f^t(\gamma_o) \leq -\epsilon) \leq e^{-\frac{2|D|\epsilon^2}{(ub-lb)^2}}, \quad \forall \epsilon > 0.$$

Let $\epsilon = f^t(\gamma_o) - f^t(\gamma_s o)$, we derive

$$P(\hat{f}^t(\gamma_o) - f^t(\gamma_o) \geq f^t(\gamma_{so}) - f^t(\gamma_o)) \geq 1 - e^{-\frac{2|D|\left(f^t(\gamma_{so}) - f^t(\gamma_o)\right)^2}{(ub-lb)^2}}.$$

$\square$

This proposition establishes a theoretical basis for bounding the probability of early stopping decision errors in terms of both the dataset size $|D|$ and expected loss variance. First, larger datasets tend to reduce discrepancies between estimates and expectations, thereby increasing the likelihood of obtaining precise measurements. Second, the magnitude of loss discrepancies between different model configurations over the overall data distribution $U$ inversely impacts the likelihood of making incorrect early stopping decisions on a given dataset $D$. The specific expectation of the loss metric on the overall distribution $U$ depends on the characteristics of the chosen loss metric function. In essence, the expectations drawn from the overall distribution $U$ are contingent on the particular loss metric function in use. A metric that effectively captures model capabilities and can distinguish variations among different model configurations is bound to result in higher reliability in early stopping decisions.

## D   EFFECT OF COMBINING TRAINING AND VALIDATION METRICS

In this part, we examine metrics that integrate both training and validation information as a foundation for early stopping. We adopt the same linear weighting approach as detailed in Section 4.2.2. Our objective is to offer insights into the potential performance enhancements achievable through this metrics integration. Specifically, we take into consideration the stability and reliability of training

information during the model's unconverged phase, as well as the role of validation information as an indicator of generalization during the later stages of model training. To achieve this, we progressively decrease the weight assigned to the training information using the formula $w_t = 1 - \frac{t}{T}$, while concurrently increasing the weight assigned to the validation information with $w_v = 1 - w_t$. We denote the metrics that combine both training and validation losses as $C_{loss}$, and metrics that combine both training and validation accuracy as $C_{acc}$. We present their performance improvements in Tables 19 and 20. Our observations reveal that this novel metric, which linearly combines and weighs the contributions of both metrics, produces effects that lie between the two individual metrics. We posit that a more refined approach to combining these sources of information may lead to even more stable and superior results.

Table 19: P-value from a Wilcoxon signed-rank test for the combined metrics against commonly used metrics on Nas-Bench-201. A p-value below 0.01 indicates that metric $A$ demonstrates superior performance compared to metric $B$ in the "$A$ vs. $B$" comparison. We also give the number of wins and losses where metric $A$ outperforms or lags behind metric $B$. $R$ denotes the budget constraint of Hyperband.

| Benchmark | | $C_{loss}$ vs. $V_{loss}$ | | | | | $C_{acc}$ vs. $V_{acc}$ | | | | |
|---|---|---|---|---|---|---|---|---|---|---|---|
| | | $R=20$ | $R=50$ | $R=81$ | $R=120$ | $R=150$ | $R=20$ | $R=50$ | $R=81$ | $R=120$ | $R=150$ |
| CIFAR-10 | p-values | $7.6e^{-49}$ | $9.4e^{-85}$ | $2.7e^{-89}$ | $2.1e^{-52}$ | $3.7e^{-38}$ | $3.1e^{-37}$ | $1.4e^{-71}$ | $4.9e^{-81}$ | $9.6e^{-43}$ | $1.9e^{-29}$ |
| | wins/losses | 392/91 | 524/71 | 564/73 | 454/114 | 368/137 | 329/90 | 438/57 | 512/81 | 390/114 | 310/127 |
| CIFAR-100 | p-values | $7.9e^{-28}$ | $2.2e^{-66}$ | $1.5e^{-73}$ | $2.9e^{-63}$ | $7.8e^{-31}$ | $2.5e^{-27}$ | $5.3e^{-54}$ | $1.0e^{-54}$ | $6.6e^{-44}$ | $1.2e^{-26}$ |
| | wins/losses | 290/109 | 427/60 | 486/78 | 437/96 | 336/119 | 66/91 | 372/64 | 402/77 | 337/88 | 272/95 |
| ImageNet-12-160 | p-values | $2.7e^{-39}$ | $6.0e^{-46}$ | $1.6e^{-45}$ | $3.8e^{-19}$ | $3.0e^{-24}$ | $1.7e^{-24}$ | $3.9e^{-37}$ | $1.0e^{-34}$ | $9.5e^{-15}$ | $7.2e^{-16}$ |
| | wins/losses | 288/63 | 314/48 | 343/77 | 236/101 | 239/88 | 220/55 | 276/55 | 289/80 | 196/87 | 194/97 |

Table 20: P-value from a Wilcoxon signed-rank test for the combined metrics against commonly used metrics on LCBench. A p-value below 0.01 indicates that metric $A$ demonstrates superior performance compared to metric $B$ in the "$A$ vs. $B$" comparison. We also give the number of wins and losses where metric $A$ outperforms or lags behind metric $B$. $R$ denotes the budget constraint of Hyperband.

| Benchmark | | $C_{loss}$ vs. $T_{loss}$ | | | | | $C_{acc}$ vs. $T_{acc}$ | | | | |
|---|---|---|---|---|---|---|---|---|---|---|---|
| | | $R=3$ | $R=6$ | $R=10$ | $R=15$ | $R=30$ | $R=3$ | $R=6$ | $R=10$ | $R=15$ | $R=30$ |
| Fashion-MNIST | p-values | $1.1e^{-3}$ | 0.045 | $1.3e^{-6}$ | $1.4e^{-6}$ | $2.2e^{-16}$ | 0.5 | 0.03 | $3.3e^{-5}$ | $2.6e^{-5}$ | $1.8e^{-17}$ |
| | wins/losses | 12/0 | 6/1 | 33/5 | 42/4 | 107/19 | 2/2 | 6/1 | 29/8 | 49/15 | 123/25 |
| adult | p-values | 0.22 | 0.26 | 0.51 | 0.96 | 0.99 | 0.47 | 0.1 | 0.98 | 0.98 | 0.99 |
| | wins/losses | 6/3 | 6/3 | 33/32 | 35/50 | 66/119 | 9/11 | 22/18 | 326/131 | 66/90 | 99/215 |
| higgs | p-values | 0.022 | $8.6e^{-3}$ | $7.8e^{-5}$ | 0.021 | 0.071 | 0.18 | 0.026 | $6.2e^{-15}$ | $6.9e^{-4}$ | $3.5e^{-8}$ |
| | wins/losses | 5/0 | 7/1 | 24/4 | 15/10 | 17/8 | 7/7 | 11/7 | 18/7 | 25/15 | 49/7 |
| jasmine | p-values | 0.062 | 0.18 | 0.44 | 0.99 | $7.2e^{-51}$ | 0.20 | 0.10 | 0.32 | 0.037 | $2.5e^{-40}$ |
| | wins/losses | 5/3 | 6/4 | 27/19 | 30/56 | 340/69 | 6/7 | 10/5 | 25/17 | 47/29 | 241/26 |
| vehicle | p-values | $2.6e^{-3}$ | 0.022 | 0.014 | 0.45 | 0.069 | $4.5e^{-9}$ | $1.4e^{-3}$ | $2.8e^{-3}$ | $1.4e^{-3}$ | $4.1e^{-34}$ |
| | wins/losses | 16/4 | 12/5 | 12/6 | 11/10 | 31/22 | 30/3 | 11/1 | 20/9 | 45/26 | 245/31 |
| volkert | p-values | 0.072 | $1.4e^{-3}$ | $6.7e^{-6}$ | $8.8e^{-8}$ | $6.8e^{-12}$ | 0.70 | 0.91 | $3.7e^{-3}$ | $9.3e^{-4}$ | $2.3e^{-5}$ |
| | wins/losses | 3/1 | 12/2 | 35/9 | 50/12 | 80/10 | 1/2 | 5/10 | 32/16 | 56/36 | 65/29 |

# E    MORE RESULTS ON THE INTEGRATION OF UNCERTAINTY

## E.1    TECHNICAL DETAILS

In this set of experiments, we test the impact of performance smoothing and ensemble averaging metrics derived from training and validation losses on HPO results. The experimental setup aligns with Appendix A.

Specifically, the performance smoothing metrics experiments encompass a range of window sizes (2 to 6), probing potential improvements across varying task complexities. Figure 5 shows the discrepancy between HPO outcomes utilizing different window sizes for calculating performance smoothing metrics and the original empirical losses. Cohen's d, described in Eq. A.1, quantifies this gap across five benchmarks with distinct budget settings. The dot shape denotes the Wilcoxon signed-rank test's significance, showcasing the difference between performance smoothing metrics and original empirical losses; the absence of dots signifies an insignificant difference. A positive Cohen's d implies the superiority of performance smoothing metrics over original empirical losses.

In evaluating ensemble averaging, we aggregate results from three runs with distinct random seeds. Once again, the Wilcoxon signed-rank test validates the significance of differences between ensemble averaging metrics and the original empirical losses, as illustrated in Figure 6.

## E.2    PERFORMANCE SMOOTHING

We present the results of applying performance smoothing metrics to both training loss and validation loss. These experiments include various window sizes and are conducted across different benchmarks while adhering to distinct budget constraints. We collect results for Hyperband in Tables 21 and 22, BOHB in Tables 23 and 24, and SS in Tables 25 and 26. Our findings within the Nas-Bench-201 benchmark reveal that the performance smoothing metrics yield only a marginal improvement when compared to training loss, with nearly 95% of repetitions resulting in a tie. However, a noteworthy enhancement is observed when we contrast it with validation loss, with the most substantial gains occurring around window sizes 5 and 6. In contrast, for LCBench, the performance smoothing metrics exhibit minimal effectiveness, and its performance diminishes as the window size increases. This phenomenon can be attributed to the rapid convergence of models in LCBench, leading to significant fluctuations between neighboring epochs. Consequently, the foundational assumption of the performance smoothing metrics, which posits that variations in successive epochs primarily reflect uncertainties, is not met in this particular context.

Table 21: P-value from a Wilcoxon signed-rank test for the performance smoothing metrics against commonly used metrics on Nas-Bench-201. A p-value below $0.01$ indicates that metric $A$ demonstrates superior performance compared to metric $B$ in the "$A$ vs. $B$" comparison. We also give the number of wins and losses where metric $A$ outperforms or lags behind metric $B$. $R$ denotes the budget constraint of Hyperband, and # window denotes the window size.

| Benchmark | # window | | $C_\mu$ vs. $T_{loss}$ | | | | | $C_\mu$ vs. $V_{loss}$ | | | | |
|---|---|---|---|---|---|---|---|---|---|---|---|---|
| | | | $R=20$ | $R=50$ | $R=81$ | $R=120$ | $R=150$ | $R=20$ | $R=50$ | $R=81$ | $R=120$ | $R=150$ |
| CIFAR-10 | 2 | p-values | 0.93 | 0.32 | 0.11 | $1.1e^{-5}$ | 0.13 | 0.31 | 0.001 | $9.4e^{-4}$ | 0.015 | $1.2e^{-4}$ |
| | | wins/losses | 10/22 | 34/32 | 72/57 | 112/58 | 109/87 | 199/189 | 328/271 | 373/312 | 370/310 | 323/279 |
| | 4 | p-values | 0.99 | 0.28 | 0.54 | $1.3e^{-6}$ | $7.9e^{-3}$ | 0.04 | $4.3e^{-11}$ | $1.7e^{-12}$ | $9.1e^{-15}$ | $2.3e^{-11}$ |
| | | wins/losses | 30/45 | 44/40 | 69/67 | 140/82 | 157/118 | 264/211 | 392/279 | 460/311 | 447/289 | 400/269 |
| | 6 | p-values | 0.99 | 0.65 | 0.64 | $5.3e^{-7}$ | $2.2e^{-5}$ | $8.4e^{-4}$ | $1.8e^{-16}$ | $1.4e^{-25}$ | $1.2e^{-18}$ | $2.3e^{-23}$ |
| | | wins/losses | 39/69 | 43/47 | 73/75 | 147/82 | 184/116 | 282/201 | 417/274 | 523/273 | 488/288 | 460/251 |
| CIFAR-100 | 2 | p-values | 0.99 | 0.97 | 0.22 | 0.072 | 0.049 | 0.035 | $3.2e^{-7}$ | $2.5e^{-3}$ | $3.2e^{-13}$ | 0.013 |
| | | wins/losses | 5/19 | 12/12 | 23/18 | 24/10 | 21/19 | 187/150 | 259/182 | 312/286 | 367/227 | 335/273 |
| | 4 | p-values | 0.99 | 0.99 | 0.037 | 0.29 | 0.25 | 0.089 | $1.2e^{-13}$ | $3.0e^{-9}$ | $1.8e^{-28}$ | $1.1e^{-19}$ |
| | | wins/losses | 15/40 | 14/28 | 34/25 | 30/27 | 28/27 | 215/191 | 324/181 | 386/279 | 469/247 | 454/252 |
| | 6 | p-values | 0.99 | 0.99 | 0.19 | 0.09 | 0.15 | 0.05 | $3.2e^{-15}$ | $7.6e^{-17}$ | $2.1e^{-32}$ | $2.0e^{-21}$ |
| | | wins/losses | 18/69 | 16/34 | 34/30 | 33/28 | 33/28 | 223/194 | 311/205 | 426/261 | 501/242 | 471/232 |
| ImageNet-16-120 | 2 | p-values | 0.99 | 0.43 | 0.51 | 0.56 | 0.39 | 0.17 | $1.6e^{-6}$ | $8.2e^{-10}$ | $4.7e^{-4}$ | $8.1e^{-8}$ |
| | | wins/losses | 3/11 | 7/6 | 12/12 | 7/7 | 11/8 | 152/128 | 208/141 | 280/177 | 241/191 | 232/148 |
| | 4 | p-values | 0.99 | 0.77 | 0.69 | 0.5 | 0.77 | $3.3e^{-3}$ | $3.3e^{-12}$ | $3.9e^{-22}$ | $5.7e^{-6}$ | $9.4e^{-11}$ |
| | | wins/losses | 6/9 | 9/12 | 18/19 | 7/5 | 9/11 | 191/120 | 261/144 | 340/171 | 273/199 | 298/173 |
| | 6 | p-values | 0.99 | 0.99 | 0.64 | 0.36 | 0.79 | 0.05 | $9.7e^{-16}$ | $2.2e^{-28}$ | $3.8e^{-8}$ | $1.4e^{-14}$ |
| | | wins/losses | 13/15 | 9/18 | 15/17 | 6/5 | 11/12 | 185/135 | 290/139 | 381/182 | 70/191 | 299/167 |

Table 22: P-value from a Wilcoxon signed-rank test for the performance smoothing metrics against commonly used metrics on LCBench. A p-value below $0.01$ indicates that metric $A$ demonstrates superior performance compared to metric $B$ in the "$A$ vs. $B$" comparison. We also give the number of wins and losses where metric $A$ outperforms or lags behind metric $B$. $R$ denotes the budget constraint of Hyperband, and # window denotes the window size.

| Benchmark | # window | | $C_\mu$ vs. $T_{loss}$ | | | | | $C_\mu$ vs. $V_{loss}$ | | | | |
|---|---|---|---|---|---|---|---|---|---|---|---|---|
| | | | $R=20$ | $R=50$ | $R=81$ | $R=120$ | $R=150$ | $R=20$ | $R=50$ | $R=81$ | $R=120$ | $R=150$ |
| Fashion-MNIST | 2 | p-values | 0.99 | 0.99 | 0.85 | 0.99 | 0.89 | 0.99 | 0.99 | 0.94 | 0.99 | 0.78 |
| | | wins/losses | 53/76 | 10/51 | 46/71 | 4/50 | 36/43 | 57/101 | 13/47 | 61/78 | 19/50 | 44/58 |
| | 4 | p-values | 0.99 | 0.99 | 0.99 | 0.99 | 0.67 | 1.0 | 1.0 | 1.0 | 1.0 | 1.0 |
| | | wins/losses | 83/179 | 32/80 | 113/157 | 27/98 | 52/64 | 66/198 | 37/119 | 100/151 | 18/95 | 53/60 |
| | 6 | p-values | 0.99 | 0.99 | 0.99 | 0.99 | 0.93 | 0.99 | 1.0 | 0.99 | 0.99 | 0.60 |
| | | wins/losses | 83/179 | 43/157 | 111/178 | 51/149 | 52/72 | 66/198 | 38/163 | 99/159 | 33/134 | 56/70 |
| adult | 2 | p-values | 0.99 | 0.99 | 0.99 | 0.99 | 0.95 | 0.99 | 0.99 | 0.023 | $2.9e^{-3}$ | $9.2e^{-5}$ |
| | | wins/losses | 44/66 | 10/44 | 26/61 | 23/51 | 38/48 | 44/91 | 32/64 | 132/107 | 99/75 | 168/112 |
| | 4 | p-values | 0.99 | 0.99 | 1.0 | 0.99 | 0.99 | 0.99 | 0.99 | 0.95 | 0.088 | $2.6e^{-7}$ |
| | | wins/losses | 70/139 | 16/83 | 36/157 | 36/97 | 37/100 | 69/142 | 35/101 | 165/194 | 128/110 | 216/126 |
| | 6 | p-values | 0.99 | 1.0 | 1.0 | 1.0 | 0.99 | 0.99 | 1.0 | 0.99 | 0.61 | $3.8e^{-8}$ |
| | | wins/losses | 28/83 | 24/64 | 36/127 | 41/142 | 36/91 | 69/142 | 38/148 | 166/200 | 132/138 | 222/132 |
| higgs | 2 | p-values | 0.99 | 0.99 | $5.8e^{-3}$ | 0.99 | $1.4e^{-6}$ | 0.99 | 0.99 | $2.0e^{-4}$ | 0.99 | $7.0e^{-12}$ |
| | | wins/losses | 99/214 | 17/65 | 109/120 | 6/29 | 74/35 | 114/220 | 10/38 | 123/116 | 28/31 | 85/26 |
| | 4 | p-values | 1.0 | 0.99 | $2.0e^{-7}$ | 0.1 | $2.0e^{-8}$ | 0.99 | 0.99 | $2.0e^{-4}$ | 0.99 | $7.0e^{-12}$ |
| | | wins/losses | 141/318 | 94/206 | 167/133 | 76/106 | 82/33 | 165/292 | 85/173 | 165/119 | 76/73 | 91/28 |
| | 6 | p-values | 1.0 | 0.99 | $2.2e^{-4}$ | $1.6e^{-5}$ | $6.6e^{-5}$ | 0.99 | 0.99 | $2.0e^{-6}$ | $1.4e^{-5}$ | $2.0e^{-11}$ |
| | | wins/losses | 141/318 | 97/242 | 164/152 | 108/103 | 73/44 | 165/292 | 91/199 | 162/128 | 106/78 | 97/34 |
| jasmine | 2 | p-values | 1.0 | 0.85 | 0.99 | 0.028 | 0.99 | 0.99 | 0.09 | 0.50 | $2.2e^{-4}$ | 0.014 |
| | | wins/losses | 27/130 | 23/25 | 61/93 | 35/21 | 41/58 | 30/93 | 20/16 | 75/69 | 68/34 | 64/38 |
| | 4 | p-values | 1.0 | 0.99 | 0.99 | $3.9e^{-9}$ | 0.79 | 0.99 | 0.74 | 0.99 | $1.8e^{-7}$ | 0.14 |
| | | wins/losses | 49/176 | 56/84 | 79/117 | 90/34 | 57/56 | 43/133 | 74/99 | 157/220 | 64/183 | 607/168 |
| | 6 | p-values | 1.0 | 0.97 | 0.92 | $9.8e^{-6}$ | 0.32 | 0.99 | 0.25 | 0.98 | $4.3e^{-4}$ | 0.18 |
| | | wins/losses | 49/176 | 80/92 | 103/122 | 94/50 | 68/55 | 43/133 | 67/59 | 113/138 | 145/94 | 107/92 |
| vehicle | 2 | p-values | 0.99 | 0.99 | 0.99 | 0.99 | 0.36 | 0.99 | 0.99 | 0.99 | 0.92 | 0.95 |
| | | wins/losses | 22/71 | 12/28 | 17/41 | 10/21 | 28/30 | 9/80 | 2/17 | 13/24 | 6/12 | 10/14 |
| | 4 | p-values | 1.0 | 0.99 | 0.99 | 0.99 | 0.42 | 0.99 | 0.99 | 0.99 | 0.92 | 0.95 |
| | | wins/losses | 25/130 | 16/48 | 23/61 | 20/39 | 47/47 | 16/117 | 5/46 | 24/34 | 29/28 | 15/26 |
| | 6 | p-values | 1.0 | 0.99 | 0.99 | 0.99 | 0.56 | 1.0 | 0.99 | 0.99 | 0.86 | 0.97 |
| | | wins/losses | 25/130 | 16/59 | 27/68 | 23/46 | 55/59 | 16/117 | 6/56 | 32/44 | 46/48 | 25/41 |
| volkert | 2 | p-values | 0.98 | 0.99 | 0.99 | 0.99 | 0.99 | 0.99 | 0.91 | 0.77 | 0.79 | 0.16 |
| | | wins/losses | 29/42 | 5/35 | 27/46 | 1/28 | 13/45 | 37/52 | 15/27 | 71/83 | 51/45 | 64/59 |
| | 4 | p-values | 0.99 | 0.99 | 0.99 | 0.99 | 0.99 | 0.99 | 0.91 | 0.77 | 0.79 | 0.16 |
| | | wins/losses | 40/81 | 20/56 | 46/106 | 12/88 | 25/95 | 42/72 | 30/56 | 72/99 | 57/93 | 61/87 |
| | 6 | p-values | 0.99 | 0.99 | 1.0 | 0.99 | 0.99 | 0.99 | 0.91 | 0.77 | 0.79 | 0.16 |
| | | wins/losses | 40/81 | 33/110 | 45/129 | 25/124 | 24/110 | 42/72 | 39/96 | 77/111 | 70/117 | 58/99 |

Table 23: P-value from a Wilcoxon signed-rank test for the performance smoothing metrics against commonly used metrics on Nas-Bench-201. A p-value below $0.01$ indicates that metric $A$ demonstrates superior performance compared to metric $B$ in the "$A$ vs. $B$" comparison. We also give the number of wins and losses where metric $A$ outperforms or lags behind metric $B$. $R$ denotes the budget constraint of **BOHB**, and # window denotes the window size.

| Benchmark | # window | | $C_\mu$ vs. $T_{loss}$ | | | | | $C_\mu$ vs. $V_{loss}$ | | | | |
|---|---|---|---|---|---|---|---|---|---|---|---|---|
| | | | $R=20$ | $R=50$ | $R=81$ | $R=120$ | $R=150$ | $R=20$ | $R=50$ | $R=81$ | $R=120$ | $R=150$ |
| CIFAR-10 | 2 | p-values | 0.60 | 0.09 | 0.25 | 0.36 | 0.46 | 0.072 | 0.081 | $4.7e^{-4}$ | 0.13 | 0.02 |
| | | wins/losses | 502/494 | 512/482 | 511/480 | 482/506 | 506/482 | 530/467 | 507/487 | 533/460 | 514/482 | 531/463 |
| | 4 | p-values | 0.76 | 0.14 | 0.21 | 0.28 | 0.42 | $4.5e^{-3}$ | $4.8e^{-7}$ | $9.4e^{-11}$ | $8.1e^{-10}$ | $1.5e^{-6}$ |
| | | wins/losses | 497/500 | 511/483 | 508/484 | 492/493 | 509/481 | 549/449 | 554/441 | 584/407 | 562/432 | 561/434 |
| | 6 | p-values | 0.90 | 0.17 | 0.22 | 0.44 | 0.34 | 0.02 | $7.3e^{-9}$ | $1.2e^{-15}$ | $2.3e^{-13}$ | $2.5e^{-11}$ |
| | | wins/losses | 490/506 | 512/483 | 506/486 | 487/499 | 508/480 | 535/464 | 560/435 | 602/389 | 585/411 | 583/412 |
| CIFAR-100 | 2 | p-values | 0.91 | 0.36 | 0.063 | 0.37 | 0.054 | 0.41 | $3.9e^{-4}$ | 0.10 | $1.7e^{-4}$ | 0.68 |
| | | wins/losses | 473/527 | 501/497 | 524/475 | 506/493 | 513/485 | 499/501 | 544/455 | 519/479 | 554/444 | 504/493 |
| | 4 | p-values | 0.96 | 0.33 | 0.061 | 0.41 | 0.076 | 0.14 | $2.1e^{-7}$ | $4.7e^{-4}$ | $1.2e^{-12}$ | $9.3e^{-5}$ |
| | | wins/losses | 468/532 | 504/494 | 524/475 | 502/497 | 512/486 | 508/492 | 568/432 | 542/456 | 596/401 | 557/441 |
| | 6 | p-values | 0.99 | 0.43 | 0.071 | 0.34 | 0.079 | 0.41 | $3.9e^{-4}$ | 0.10 | $1.7e^{-4}$ | 0.68 |
| | | wins/losses | 462/538 | 498/500 | 526/473 | 506/493 | 512/486 | 519/481 | 560/440 | 572/427 | 618/379 | 587/413 |
| ImageNet-16-120 | 2 | p-values | 0.94 | 0.77 | 0.09 | 0.77 | 0.62 | 0.18 | 0.012 | 0.011 | 0.22 | 0.22 |
| | | wins/losses | 518/481 | 500/498 | 480/519 | 510/487 | 506/492 | 503/496 | 536/464 | 524/476 | 503/495 | 502/495 |
| | 4 | p-values | 0.96 | 0.78 | 0.13 | 0.81 | 0.61 | 0.16 | $6.0e^{-5}$ | $3.1e^{-8}$ | 0.17 | 0.015 |
| | | wins/losses | 517/482 | 497/501 | 515/484 | 485/512 | 494/504 | 504/495 | 549/451 | 558/439 | 504/494 | 520/477 |
| | 6 | p-values | 0.98 | 0.81 | 0.11 | 0.82 | 0.71 | 0.14 | $1.3e^{-6}$ | $1.0e^{-8}$ | 0.056 | 0.003 |
| | | wins/losses | 481/518 | 496/502 | 488/509 | 491/507 | 504/495 | 567/433 | 562/435 | 507/491 | 538/459 | 460/251 |

Table 24: P-value from a Wilcoxon signed-rank test for the performance smoothing metrics against commonly used metrics on LCBench. A p-value below $0.01$ indicates that metric $A$ demonstrates superior performance compared to metric $B$ in the "$A$ vs. $B$" comparison. We also give the number of wins and losses where metric $A$ outperforms or lags behind metric $B$. $R$ denotes the budget constraint of **_BOHB_**, and # window denotes the window size.

| Benchmark | # window | | $C_\mu$ vs. $T_{loss}$ | | | | | $C_\mu$ vs. $V_{loss}$ | | | | |
|---|---|---|---|---|---|---|---|---|---|---|---|---|
| | | | $R=20$ | $R=50$ | $R=81$ | $R=120$ | $R=150$ | $R=20$ | $R=50$ | $R=81$ | $R=120$ | $R=150$ |
| Fashion-MNIST | 2 | p-values | 0.99 | 0.99 | 0.95 | 0.99 | 0.93 | 0.99 | 0.99 | 0.99 | 0.99 | 0.98 |
| | | wins/losses | 57/95 | 6/56 | 39/66 | 7/53 | 34/48 | 56/125 | 16/49 | 47/85 | 21/49 | 40/63 |
| | 4 | p-values | 0.99 | 0.99 | 0.99 | 0.99 | 0.99 | 1.0 | 1.0 | 1.0 | 1.0 | 0.94 |
| | | wins/losses | 87/188 | 35/104 | 88/158 | 13/112 | 50/75 | 68/217 | 40/116 | 83/179 | 37/101 | 45/60 |
| | 6 | p-values | 0.99 | 0.99 | 0.99 | 0.99 | 1.0 | 0.99 | 1.0 | 0.99 | 0.99 | 0.63 |
| | | wins/losses | 87/188 | 47/172 | 91/168 | 36/176 | 46/93 | 68/217 | 41/158 | 82/188 | 57/140 | 57/65 |
| adult | 2 | p-values | 0.99 | 0.99 | 0.99 | 0.99 | 0.078 | 0.99 | 0.99 | 0.032 | 0.056 | 0.95 |
| | | wins/losses | 52/73 | 15/52 | 26/55 | 11/30 | 62/51 | 56/85 | 49/60 | 117/93 | 99/88 | 96/117 |
| | 4 | p-values | 0.99 | 0.99 | 1.0 | 0.99 | 0.89 | 0.99 | 0.99 | 0.95 | 0.53 | 0.32 |
| | | wins/losses | 68/136 | 20/90 | 36/157 | 22/79 | 59/67 | 70/153 | 48/106 | 138/163 | 124/118 | 127/123 |
| | 6 | p-values | 0.99 | 1.0 | 1.0 | 1.0 | 0.92 | 0.99 | 0.99 | 0.90 | 0.97 | 0.16 |
| | | wins/losses | 68/136 | 27/153 | 38/176 | 34/156 | 68/79 | 70/153 | 53/162 | 141/160 | 141/158 | 140/114 |
| higgs | 2 | p-values | 0.99 | 0.99 | 0.27 | 0.99 | 0.72 | 0.99 | 0.99 | $2.0e^{-3}$ | 0.99 | 0.58 |
| | | wins/losses | 115/178 | 25/48 | 104/145 | 15/37 | 43/41 | 128/201 | 9/40 | 110/116 | 26/38 | 43/41 |
| | 4 | p-values | 0.99 | 0.99 | 0.026 | 0.40 | 0.68 | 0.99 | 0.99 | $2.3e^{-3}$ | 0.092 | 0.25 |
| | | wins/losses | 161/285 | 115/187 | 141/169 | 85/112 | 59/74 | 174/280 | 99/164 | 139/148 | 88/84 | 58/57 |
| | 6 | p-values | 0.99 | 0.99 | 0.23 | $4.4e^{-3}$ | 0.84 | 0.99 | 0.99 | 0.073 | $8.4e^{-4}$ | 0.84 |
| | | wins/losses | 161/285 | 125/226 | 139/184 | 105/110 | 64/85 | 174/280 | 100/184 | 134/160 | 113/86 | 57/76 |
| jasmine | 2 | p-values | 1.0 | 0.99 | 0.99 | $2.8e^{-4}$ | 0.69 | 0.99 | 0.90 | 0.13 | 0.066 | 0.98 |
| | | wins/losses | 30/140 | 16/31 | 59/96 | 31/12 | 48/52 | 36/99 | 20/25 | 87/70 | 62/54 | 69/80 |
| | 4 | p-values | 1.0 | 0.99 | 0.99 | $3.9e^{-9}$ | 0.79 | 0.99 | 0.74 | 0.99 | $1.8e^{-7}$ | 0.14 |
| | | wins/losses | 48/183 | 47/78 | 81/127 | 87/26 | 55/53 | 72/216 | 71/117 | 222/235 | 282/264 | 242/487 |
| | 6 | p-values | 1.0 | 0.99 | 0.99 | $3.2e^{-10}$ | 0.49 | 0.99 | 0.99 | 0.99 | $3.2e^{-3}$ | 0.29 |
| | | wins/losses | 48/183 | 57/81 | 101/133 | 91/40 | 87/65 | 48/137 | 61/78 | 117/142 | 122/105 | 129/116 |
| vehicle | 2 | p-values | 0.99 | 0.99 | 0.99 | 0.96 | 0.57 | 0.99 | 0.99 | 0.98 | 0.97 | 0.33 |
| | | wins/losses | 23/91 | 11/41 | 23/36 | 10/13 | 39/36 | 11/62 | 2/25 | 18/23 | 8/13 | 36/30 |
| | 4 | p-values | 0.99 | 0.99 | 0.99 | 0.99 | 0.62 | 0.99 | 0.99 | 0.99 | 0.97 | 0.10 |
| | | wins/losses | 27/124 | 16/76 | 30/59 | 19/38 | 50/48 | 14/100 | 6/69 | 26/36 | 18/22 | 38/38 |
| | 6 | p-values | 1.0 | 0.99 | 0.99 | 0.99 | 0.82 | 1.0 | 0.99 | 0.99 | 0.99 | 0.78 |
| | | wins/losses | 27/127 | 18/87 | 33/63 | 21/48 | 56/60 | 14/100 | 13/81 | 28/42 | 20/35 | 49/55 |
| volkert | 2 | p-values | 0.98 | 0.99 | 0.99 | 0.99 | 0.69 | 0.99 | 0.95 | 0.98 | 0.99 | 0.93 |
| | | wins/losses | 17/47 | 6/31 | 17/47 | 4/29 | 40/35 | 32/66 | 17/22 | 59/98 | 37/57 | 59/71 |
| | 4 | p-values | 0.99 | 0.99 | 0.99 | 0.99 | 0.99 | 0.99 | 0.99 | 0.99 | 0.99 | 0.86 |
| | | wins/losses | 26/96 | 19/43 | 31/113 | 13.82 | 41/61 | 30/98 | 29/48 | 64/108 | 46/99 | 67/84 |
| | 6 | p-values | 0.99 | 0.99 | 1.0 | 0.99 | 0.99 | 0.99 | 0.99 | 0.99 | 0.99 | 0.95 |
| | | wins/losses | 26/96 | 29/102 | 31/125 | 24/134 | 49/73 | 30/98 | 35/96 | 69/124 | 59/125 | 63/89 |

Table 25: P-value from a Wilcoxon signed-rank test for the performance smoothing metrics against commonly used metrics on Nas-Bench-201. A p-value below $0.01$ indicates that metric $A$ demonstrates superior performance compared to metric $B$ in the "$A$ vs. $B$" comparison. We also give the number of wins and losses where metric $A$ outperforms or lags behind metric $B$. $R$ denotes the budget constraint of $\textbf{SS}$, and $|W|$ denotes the window size.

| Benchmark | R | | $C_\mu$ vs. $T_{loss}$ | | | $C_\mu$ vs. $V_{loss}$ | | |
|---|---|---|---|---|---|---|---|---|
| | | | $|W| = 2$ | $|W| = 4$ | $|W| = 6$ | $|W| = 2$ | $|W| = 4$ | $|W| = 6$ |
| CIFAR-10 | 20 | p-values | 0.34 | $7.3e^{-5}$ | $2.5e^{-7}$ | 0.70 | 0.033 | $1.3e^{-3}$ |
| | | wins/losses | 512/484 | 554/442 | 572/424 | 511/487 | 519/479 | 534/464 |
| CIFAR-100 | 20 | p-values | 0.19 | $1.7e^{-5}$ | $3.6e^{-10}$ | 0.51 | $3.3e^{-3}$ | $1.2e^{-3}$ |
| | | wins/losses | 519/481 | 558/442 | 581/419 | 505/495 | 531/469 | 532/468 |
| ImageNet-16-120 | 20 | p-values | 0.12 | $1.7e^{-6}$ | $2.2e^{-10}$ | 0.56 | 0.41 | 0.26 |
| | | wins/losses | 518/481 | 500/498 | 519/480 | 510/487 | 506/492 | 503/496 |

Table 26: P-value from a Wilcoxon signed-rank test for the performance smoothing metrics against commonly used metrics on LCBench. A p-value below $0.01$ indicates that metric $A$ demonstrates superior performance compared to metric $B$ in the "$A$ vs. $B$" comparison. We also give the number of wins and losses where metric $A$ outperforms or lags behind metric $B$. $R$ denotes the budget constraint of $\textbf{SS}$, and # window denotes the window size.

| Benchmark | # window | | $C_\mu$ vs. $T_{loss}$ | | | | | $C_\mu$ vs. $V_{loss}$ | | | | |
|---|---|---|---|---|---|---|---|---|---|---|---|---|
| | | | $R = 20$ | $R = 50$ | $R = 81$ | $R = 120$ | $R = 150$ | $R = 20$ | $R = 50$ | $R = 81$ | $R = 120$ | $R = 150$ |
| Fashion-MNIST | 2 | p-values | 0.17 | 0.022 | 1.0 | 1.0 | 0.99 | 0.17 | 0.022 | 1.0 | 1.0 | 0.99 |
| | | wins/losses | 507/491 | 535/465 | 319/677 | 306/693 | 453/541 | 507/491 | 535/465 | 256/742 | 239/760 | 381/616 |
| adult | 2 | p-values | 0.78 | 0.15 | 1.0 | 1.0 | 0.99 | 0.78 | 0.15 | 1.0 | 1.0 | 1.0 |
| | | wins/losses | 489/507 | 494/499 | 386/609 | 380/613 | 448/542 | 490/503 | 496/497 | 307/689 | 328/665 | 383/607 |
| higgs | 2 | p-values | 0.23 | 0.30 | 0.99 | 0.99 | 0.82 | 0.22 | 0.29 | 0.99 | 0.99 | 1.0 |
| | | wins/losses | 497/480 | 487/480 | 420/578 | 390/605 | 482/512 | 497/480 | 487/480 | 423/574 | 396/598 | 374/621 |
| jasmine | 2 | p-values | 0.72 | 0.18 | 0.99 | 1.0 | 0.66 | 0.79 | 0.17 | 1.0 | 1.0 | 0.83 |
| | | wins/losses | 479/506 | 521/469 | 379/588 | 376/595 | 451/475 | 479/506 | 521/469 | 350/608 | 333/620 | 442/490 |
| vehicle | 2 | p-values | 0.84 | 0.42 | 0.99 | 0.99 | 0.98 | 0.80 | 0.41 | 0.98 | 0.99 | 0.99 |
| | | wins/losses | 465/515 | 510/479 | 441/537 | 449/536 | 435/535 | 465/515 | 510/479 | 451/524 | 423/555 | 412/540 |
| volkert | 2 | p-values | 0.77 | 0.72 | 1.0 | 1.0 | 0.94 | 0.75 | 0.70 | 1.0 | 1.0 | 0.99 |
| | | wins/losses | 493/505 | 496/504 | 349/647 | 329/671 | 480/517 | 493/505 | 496/504 | 299/698 | 268/731 | 39/607 |

### E.3 DYNAMIC WEIGHTING

We evaluate the outcomes of metrics that incorporate uncertainty from successive epochs using both fixed and dynamic weighting approaches, compared to empirical training loss and validation loss on Nas-Bench-201. The results are presented in Table 27.

Table 27: P-value from a Wilcoxon signed-rank test for the fixed and dynamic weighting metrics against *training loss* and *validation loss* on Nas-Bench-201. A p-value below $0.01$ indicates that metric $A$ demonstrates superior performance compared to metric $B$ in the "$A$ vs. $B$" comparison. We also give the number of wins and losses where metric $A$ outperforms or lags behind metric $B$. Window size is 5 in this set of experiment. $R$ denotes the budget constraint of Hyperband.

| Benchmark | metrics | | Against *training loss* | | | | | Against *validation loss* | | | | |
|---|---|---|---|---|---|---|---|---|---|---|---|---|
| | | | $R=20$ | $R=50$ | $R=81$ | $R=120$ | $R=150$ | $R=20$ | $R=50$ | $R=81$ | $R=120$ | $R=150$ |
| CIFAR-10 | $C_{\mu-\sigma}$ | p-values | 0.72 | 0.49 | 0.15 | $3.2e^{-5}$ | 0.03 | $2.9e^{-14}$ | $5.0e^{-16}$ | $1.1e^{-20}$ | $8.1e^{-29}$ | $5.2e^{21}$ |
| | | wins/losses | 39/69 | 43/47 | 73/75 | 147/82 | 184/116 | 282/201 | 417/274 | 523/273 | 488/288 | 460/251 |
| | $C_{linear}$ | p-values | 0.87 | 0.30 | 0.16 | $8.9e^{-8}$ | $5.6e^{-4}$ | $5.7e^{-17}$ | $1.4e^{-28}$ | $3.0e^{-33}$ | $1.3e^{-23}$ | $5.8e^{-24}$ |
| | | wins/losses | 16/20 | 40/36 | 68/58 | 135/73 | 166/110 | 29/148 | 446/224 | 541/238 | 484/270 | 457/238 |
| | $C_{log}$ | p-values | 0.99 | 0.26 | 0.29 | $4.5e^{-7}$ | $2.3e^{-4}$ | $9.6e^{-13}$ | $2.5e^{-27}$ | $1.6e^{-31}$ | $2.5e^{-20}$ | $4.4e^{-19}$ |
| | | wins/losses | 24/35 | 42/37 | 67/60 | 139/82 | 171/113 | 288/155 | 442/233 | 543/253 | 477/273 | 437/248 |
| | $C_{exp}$ | p-values | 0.99 | 0.30 | 0.36 | $2.0e^{-7}$ | $2.0e^{-5}$ | $8.0e^{-6}$ | $1.5e^{-19}$ | $2.2e^{-25}$ | $7.9e^{-16}$ | $9.6e^{-16}$ |
| | | wins/losses | 26/41 | 44/39 | 68/61 | 145/84 | 180/108 | 281/188 | 423/252 | 522/263 | 458/287 | 429/272 |
| CIFAR-100 | $C_{\mu-\sigma}$ | p-values | 0.99 | 0.56 | 0.16 | 0.12 | 0.27 | 0.05 | $3.2e^{-15}$ | $7.6e^{-17}$ | $2.1e^{-32}$ | $2.0e^{-21}$ |
| | | wins/losses | 6/17 | 15/13 | 22/20 | 19/14 | 20/20 | 196/156 | 341/184 | 422/228 | 509/210 | 428/258 |
| | $C_{linear}$ | p-values | 0.99 | 0.86 | 0.11 | 0.25 | 0.06 | $4.1e^{-4}$ | $2.9e^{-22}$ | $9.0e^{-23}$ | $1.1e^{-39}$ | $1.4e^{-27}$ |
| | | wins/losses | 6/20 | 14/18 | 24/22 | 23/19 | 30/23 | 196/154 | 348/164 | 427/224 | 507/221 | 474/226 |
| | $C_{log}$ | p-values | 0.99 | 0.98 | 0.073 | 0.12 | 0.054 | $1.6e^{-3}$ | $1.1e^{-18}$ | $6.4e^{-20}$ | $4.6e^{-39}$ | $6.7e^{-24}$ |
| | | wins/losses | 9/30 | 14/26 | 29/24 | 26/22 | 31/24 | 202/162 | 344/183 | 424/237 | 488/224 | 468/238 |
| | $C_{exp}$ | p-values | 0.99 | 0.99 | 0.062 | 0.15 | 0.067 | 0.029 | $5.6e^{-15}$ | $5.6e^{-19}$ | $2.8e^{-33}$ | $4.3e^{-21}$ |
| | | wins/losses | 16/47 | 14/30 | 35/27 | 40/23 | 31/25 | 213/178 | 334/191 | 430/246 | 476/240 | 461/246 |
| ImageNet-16-120 | $C_{\mu-\sigma}$ | p-values | 0.99 | 0.19 | 0.37 | 0.39 | 0.30 | $4.2e^{-8}$ | $3.3e^{-22}$ | $2.1e^{-30}$ | $1.8e^{-9}$ | $1.3e^{-19}$ |
| | | wins/losses | 3/14 | 6/4 | 15/13 | 9/6 | 5/9 | 210/119 | 288/123 | 370/159 | 270/179 | 323/158 |
| | $C_{linear}$ | p-values | 0.99 | 0.44 | 0.55 | 0.54 | 0.35 | $3.5e^{-8}$ | $6.2e^{-24}$ | $3.4e^{-32}$ | $1.4e^{-7}$ | $2.9e^{-16}$ |
| | | wins/losses | 4/18 | 6/6 | 15/13 | 6/5 | 7/6 | 211/115 | 287/114 | 363/156 | 272/191 | 304/162 |
| | $C_{log}$ | p-values | 0.99 | 0.82 | 0.68 | 0.54 | 0.37 | $1.1e^{-5}$ | $3.4e^{-19}$ | $6.8e^{-30}$ | $2.9e^{-8}$ | $1.4e^{-17}$ |
| | | wins/losses | 5/27 | 7/10 | 16/17 | 6/5 | 8/7 | 196/112 | 285/127 | 354/161 | 281/184 | 312/151 |
| | $C_{exp}$ | p-values | 0.99 | 0.78 | 0.52 | 0.54 | 0.51 | $3.5e^{-3}$ | $3.0e^{-15}$ | $6.4e^{-26}$ | $2.9e^{-7}$ | $6.3e^{-17}$ |
| | | wins/losses | 7/33 | 10/13 | 17/17 | 6/5 | 11/9 | 194/123 | 275/139 | 351/176 | 271/192 | 307/154 |

Table 28: P-value from a Wilcoxon signed-rank test for the fixed and dynamic weighting metrics against *training loss* and *validation loss* on Nas-Bench-201. A p-value below $0.01$ indicates that metric $A$ demonstrates superior performance compared to metric $B$ in the "$A$ vs. $B$" comparison. We also give the number of wins and losses where metric $A$ outperforms or lags behind metric $B$. Window size is 5 in this set of experiment. $R$ denotes the budget constraint of ***BOHB***.

| Benchmark | metrics | | Against *training loss* | | | | | Against *validation loss* | | | | |
|---|---|---|---|---|---|---|---|---|---|---|---|---|
| | | | $R=20$ | $R=50$ | $R=81$ | $R=120$ | $R=150$ | $R=20$ | $R=50$ | $R=81$ | $R=120$ | $R=150$ |
| CIFAR-10 | $C_{\mu-\sigma}$ | p-values | 0.76 | 0.31 | 0.17 | $1.4e^{-3}$ | $5.3e^{-3}$ | $5.6e^{-13}$ | $5.5e^{-14}$ | $1.5e^{-18}$ | $9.9e^{-28}$ | $4.4e^{-23}$ |
| | | wins/losses | 15/19 | 28/24 | 47/33 | 85/50 | 79/56 | 15/21 | 27/22 | 52/36 | 90/53 | 93/58 |
| | $C_{linear}$ | p-values | 0.90 | 0.21 | 0.17 | $7.7e^{-4}$ | $5.6e^{-5}$ | $5.2e^{-14}$ | $9.5e^{-25}$ | $4.9e^{-28}$ | $9.7e^{-30}$ | $1.3e^{-24}$ |
| | | wins/losses | 16/20 | 40/36 | 68/58 | 135/73 | 166/110 | 297/160 | 431/216 | 462/240 | 471/212 | 394/211 |
| | $C_{log}$ | p-values | 0.99 | 0.47 | 0.34 | $6.8e^{-3}$ | 0.083 | $1.2e^{-11}$ | $5.5e^{-18}$ | $4.8e^{-22}$ | $1.3e^{-28}$ | $3.4e^{-23}$ |
| | | wins/losses | 473/523 | 489/506 | 502/492 | 526/474 | 483/514 | 498/501 | 539/458 | 568/427 | 539/459 | 494/502 |
| | $C_{exp}$ | p-values | 0.99 | 0.65 | 0.37 | $4.3e^{-3}$ | 0.072 | $4.6e^{-6}$ | $7.2e^{-14}$ | $6.8e^{-14}$ | $2.1e^{-20}$ | $8.0e^{-19}$ |
| | | wins/losses | 458/538 | 479/516 | 500/494 | 524/476 | 481/516 | 501/497 | 524/473 | 545/452 | 516/482 | 476/520 |
| CIFAR-100 | $C_{\mu-\sigma}$ | p-values | 0.99 | 0.43 | 0.20 | 0.84 | 0.49 | $6.4e^{-4}$ | $9.5e^{-15}$ | $4.4e^{-14}$ | $9.2e^{-28}$ | $2.1e^{-12}$ |
| | | wins/losses | 6/17 | 12/10 | 31/27 | 21/27 | 16/22 | 196/156 | 327/186 | 356/225 | 422/204 | 374/237 |
| | $C_{linear}$ | p-values | 0.99 | 0.78 | 0.08 | 0.49 | 0.54 | $4.1e^{-4}$ | $1.0e^{-17}$ | $1.9e^{-15}$ | $3.9e^{-32}$ | $1.2e^{-16}$ |
| | | wins/losses | 6/20 | 11/15 | 34/26 | 22/23 | 25/28 | 196/154 | 330/173 | 359/215 | 428/195 | 400/231 |
| | $C_{log}$ | p-values | 0.99 | 0.77 | 0.23 | 0.91 | 0.60 | $3.3e^{-7}$ | $2.0e^{-15}$ | $5.7e^{-14}$ | $2.7e^{-24}$ | $7.2e^{-17}$ |
| | | wins/losses | 472/528 | 513/487 | 508/492 | 490/510 | 503/496 | 496/503 | 541/459 | 518/480 | 516/484 | 514/485 |
| | $C_{exp}$ | p-values | 0.99 | 0.68 | 0.33 | 0.15 | 0.83 | $1.0e^{-4}$ | $1.0e^{-9}$ | $2.9e^{-9}$ | $9.3e^{-23}$ | $8.8e^{-17}$ |
| | | wins/losses | 453/547 | 507/493 | 502/498 | 487/513 | 501/498 | 497/503 | 532/468 | 503/495 | 501/499 | 508/491 |
| ImageNet-16-120 | $C_{\mu-\sigma}$ | p-values | 0.96 | 0.16 | 0.52 | 0.23 | 0.74 | $4.1e^{-12}$ | $9.8e^{-27}$ | $1.1e^{-20}$ | $1.2e^{-10}$ | $2.4e^{-16}$ |
| | | wins/losses | 4/34 | 11/11 | 22/19 | 25/19 | 25/14 | 215/94 | 322/126 | 304/136 | 223/122 | 229/109 |
| | $C_{linear}$ | p-values | 0.98 | 0.062 | 0.61 | 0.059 | 0.37 | $6.0e^{-11}$ | $1.7e^{-26}$ | $3.0e^{-22}$ | $3.0e^{-7}$ | $1.4e^{-12}$ |
| | | wins/losses | 1/6 | 12/5 | 19/17 | 3/14 | 11/11 | 218/97 | 314/120 | 303/127 | 214/136 | 229/124 |
| | $C_{log}$ | p-values | 0.99 | 0.99 | 0.87 | 0.83 | 0.35 | $1.1e^{-8}$ | $1.2e^{-14}$ | $3.2e^{-23}$ | $9.3e^{-8}$ | $1.6e^{-6}$ |
| | | wins/losses | 464/536 | 497/502 | 495/503 | 487/513 | 506/493 | 503/497 | 512/486 | 544/454 | 518/482 | 547/452 |
| | $C_{exp}$ | p-values | 0.99 | 0.99 | 0.58 | 0.65 | 0.70 | $2.3e^{-4}$ | $6.9e^{-12}$ | $3.2e^{-16}$ | $5.2e^{-7}$ | $8.8e^{-5}$ |
| | | wins/losses | 446/554 | 489/510 | 481/518 | 487/513 | 507/492 | 499/501 | 510/488 | 525/473 | 504/496 | 532/467 |

Table 29: P-value from a Wilcoxon signed-rank test for the fixed and dynamic weighting metrics against *training loss* and *validation loss* on Nas-Bench-201. A p-value below $0.01$ indicates that metric $A$ demonstrates superior performance compared to metric $B$ in the "$A$ vs. $B$" comparison. We also give the number of wins and losses where metric $A$ outperforms or lags behind metric $B$. Window size is 5 in this set of experiment. $R$ denotes the budget constraint of **SS**.

| Benchmark | metrics | | Against *training loss* | | | | Against *validation loss* | | | |
|---|---|---|---|---|---|---|---|---|---|---|
| | | | $R=20$ | $R=50$ | $R=81$ | $R=120$ | $R=20$ | $R=50$ | $R=81$ | $R=120$ |
| CIFAR-10 | $C_{\mu-\sigma}$ | p-values | 0.99 | 0.99 | 0.99 | 0.31 | 0.98 | 0.30 | 0.017 | $9.9e^{-4}$ |
| | | wins/losses | 488/508 | 489/509 | 507/487 | 527/473 | 501/498 | 508/490 | 533/462 | 526/471 |
| | $C_{linear}$ | p-values | 0.99 | 0.99 | 0.99 | 0.22 | 0.87 | $3.0e^{-5}$ | $7.2e^{-9}$ | $1.8e^{-14}$ |
| | | wins/losses | 485/511 | 488/507 | 506/488 | 527/473 | 506/493 | 539/459 | 564/431 | 567/429 |
| | $C_{log}$ | p-values | 0.99 | 0.99 | 0.99 | 0.72 | 0.98 | $6.2e^{-5}$ | $1.1e^{-7}$ | $6.6e^{-6}$ |
| | | wins/losses | 473/523 | 489/506 | 502/492 | 526/474 | 498/501 | 539/458 | 568/427 | 539/459 |
| | $C_{exp}$ | p-values | 0.99 | 0.99 | 0.87 | 0.52 | 0.026 | 0.044 | $4.4e^{-3}$ | 0.26 |
| | | wins/losses | 524/473 | 479/516 | 500/494 | 524/476 | 532/465 | 524/473 | 545/452 | 516/482 |
| CIFAR-100 | $C_{\mu-\sigma}$ | p-values | 0.99 | 0.99 | 0.99 | 0.99 | $9.5e^{-15}$ | $4.4e^{-14}$ | $9.2e^{-28}$ | $2.1e^{-12}$ |
| | | wins/losses | 482/518 | 512/479 | 521/479 | 492/508 | 491/508 | 524/476 | 502/496 | 500/500 |
| | $C_{linear}$ | p-values | 0.99 | 0.99 | 0.99 | 0.95 | 0.89 | 0.67 | 0.22 | 0.082 |
| | | wins/losses | 482/518 | 522/478 | 522/478 | 496/504 | 491/508 | 540/460 | 519/479 | 538/462 |
| | $C_{log}$ | p-values | 0.99 | 0.99 | 0.99 | 0.99 | 0.80 | 0.065 | $6.0e^{-4}$ | $2.5e^{-4}$ |
| | | wins/losses | 472/528 | 513/487 | 508/492 | 490/510 | 528/472 | 576/424 | 544/456 | 528/472 |
| | $C_{exp}$ | p-values | 0.99 | 0.99 | 0.99 | 0.99 | $4.5e^{-3}$ | 0.28 | 0.05 | 0.096 |
| | | wins/losses | 453/547 | 507/493 | 502/498 | 487/513 | 533/467 | 532/468 | 503/495 | 501/499 |
| ImageNet-16-120 | $C_{\mu-\sigma}$ | p-values | 0.99 | 0.99 | 0.65 | 0.79 | 0.067 | 0.11 | 0.017 | 0.27 |
| | | wins/losses | 486/514 | 505/494 | 502/496 | 500/500 | 502/498 | 506/492 | 522/476 | 506/494 |
| | $C_{linear}$ | p-values | 0.99 | 0.99 | 0.85 | 0.92 | 0.028 | 0.095 | $8.0e^{-7}$ | $9.5e^{-8}$ |
| | | wins/losses | 485/515 | 505/494 | 502/496 | 500/500 | 481/519 | 485/514 | 510/487 | 526/474 |
| | $C_{log}$ | p-values | 0.99 | 0.99 | 0.99 | 0.99 | 0.06 | 0.077 | $3.6e^{-5}$ | $6.0e^{-3}$ |
| | | wins/losses | 464/536 | 497/502 | 495/503 | 487/513 | 480/520 | 488/511 | 504/493 | 507/493 |
| | $C_{exp}$ | p-values | 0.99 | 0.99 | 0.99 | 0.99 | 0.14 | 0.33 | $6.7e^{-5}$ | 0.50 |
| | | wins/losses | 507/493 | 489/510 | 481/518 | 487/513 | 507/493 | 512/486 | 544/454 | 506/494 |

