# OpenReview forum: "A SYSTEMATIC STUDY ON EARLY STOPPING CRITERIA IN HPO AND THE IMPLICATIONS OF UNCERTAINTY"
_ICLR.cc/2024/Conference — Submitted to ICLR 2024_

### Official Review · Reviewer_wZMM · 2023-10-31

**Soundness:** 2 fair
**Presentation:** 1 poor
**Contribution:** 1 poor
**Rating:** 3
**Confidence:** 3

**Summary:**

The paper presents a study on early stopping criteria in hyperparameter optimization (HPO), first empirically studying the effects of budget constraints and filtering ratios on the popular Hyperband algorithm to evaluate the reliability of early stopping, and then studying the effects of model uncertainty on early stopping.

**Strengths:**

The effects of budget constraints and filtering ratios on the popular Hyperband algorithm were studied empirically using two benchmarks of varying scales, LCBench and Nas-Bench-201.

**Weaknesses:**

1.	More descriptions should be provided for the three fundamental questions to be studied in this paper. To be specific, in the absence of a description of what reliability and model uncertainty are in this paper, the motivation for this paper to evaluate the reliability of early stopping and study the model uncertainty is unclear and confusing. The second question uses words like the nature of early stopping that are not well defined, making it difficult to understand.
2.	Only the Hyperband algorithm, which was published six years ago, was examined in evaluating the utility of early stopping criteria in hyperparameter optimization. More state-of-the-art early stopping criteria should be investigated and analyzed to draw a convincing conclusion.
3.	The word gaps in Insight 2 should be described clearly.
4.	What is the fraction of budget shown in the figures? What is its relation with the filtering ratios?

**Questions:**

What are the answers to the three fundamental questions given in the Introduction?

---

> ### Author Response · Authors · 2023-11-20
> **Response to Reviewer wZMM (Part 1/2)**
>
> We sincerely thank Reviewer wZMM for the valuable suggestions. Below we would like to give detailed responses to each of your comments.
>
> **Fundamental questions and Motivation**
>
> We use the concept of reliability to assess if a specific performance metric shows statistically significant superiority over others. As we explained in Section 4.1, model uncertainty is a technical term that refers to a lack of precise knowledge or confidence in a model's predictions, typically manifested as variations in predicted outcomes.
>
> The nature of early stopping revolves around the idea of monitoring a model's performance during the training phase and halting the training process when certain criteria or performance metrics are met. This paper aims to explore the impact of using different performance metrics in the early stopping mechanisms on the final results. We conducted extensive experiments to explore the commonly used metrics - training/validation loss/accuracy - and found that training loss significantly outperforms the other metrics, especially for complex tasks with slow convergence. To this end, we showed through experiments and analysis that uncertainty and data size are possible reasons for the poor performance of validation metrics. Further, we explored a series of strategies that utilize uncertainty to help metrics improve reliability.
>
> Thanks for the suggestion. Following your advice, we have added explanation in Section 1.
>
> **Answers to the fundamental questions**
>
> This paper addresses the three fundamental questions across Sections 2 to 4, providing distinct insights into each:
>
> 1. Section 2 empirically compares commonly used criteria. It reveals that different criteria may have significantly different impacts on the HPO performance, and training loss tends to suit complex tasks better than validation loss.
>
> 2. Section 3 analyzes the factors that influence early stopping, offering insights for Section 2's findings. The nature of early stopping revolves around the idea of monitoring a model's performance during training and halting the training process when certain criteria or performance metrics are met. This practice essentially involves “a trade-off between the HPO cost and the effectiveness of the final models”. Criteria or metrics are responsible for estimating and ranking the models' capabilities. According to Section 3, the factors that affect the validity of a criterion are the size of the dataset on which the calculation is based and whether or not the criterion contains uncertainty that affects discrimination. As we analyzed in Section 3, “while a larger training dataset may contribute, the primary justification lies in the training loss serving as a superior indicator of model expressiveness, yielding greater stability and consistent model ranking. In contrast, the validation loss prioritizes the aspect of generalization and tends to favor models with lower capabilities that converge early. ”
>
> 3. Section 4 investigates the impact of model uncertainty on early stopping. Besides the commonly considered measures, other early stopping criteria include the combination of training and validation metrics validated in Section 3 and Appendix D, and a collection of methods that incorporate uncertainty explored in Section 4. Model uncertainty has some important impact on the effectiveness of early stopping. As stated in Section 4.1, “if an early stopping decision is made at a point of peak uncertainty, stronger models may be prematurely terminated.” By examining a series of criteria that introduce uncertainty, this paper concludes that “a careful combination of model uncertainty with conventional early stopping criteria can yield significant benefits for HPO, especially when dynamic trends of the training process are considered.”
>
> Thanks for raising this question. We have given a summary of our findings near the Introduction for readers to grasp the message.

---

> ### Author Response · Authors · 2023-11-20
> **Response to Reviewer wZMM (Part 2/2)**
>
> **New HPO algorithm**
>
> Thanks for the suggestion. We further conduct experiments with two widely recognized algorithms: BOHB [1] and the state-of-the-art Sub-sampling [2]. The results and detailed settings can also be found in Appendices B and E in the updated version of our paper. It can be seen that the training loss significantly outperforms other metrics across all HPO algorithms. Furthermore, introducing uncertainty into metrics demonstrates enhanced reliability—an aspect we delve into within our findings.
>
> [1] https://arxiv.org/abs/1807.01774
>
> [2] https://arxiv.org/abs/2209.12499
>
> **Terminology**
>
> Thanks for raising this concern. We clarify the terminology used in our work as follows:
>
> 1. "Gaps" in Insight 2 refers to the differences observed in performance metrics across various models, essentially reflecting the concept of performance regret. A more discriminative metric leads to clearer differences between model performances, thereby enhancing reliability.
>
> 2. The lower plots in Figure 1 depict the discrepancy between the final test accuracy achievable by the remaining model candidates after each instance of early stopping and the best final test accuracy achievable by the whole set of samples as training progresses. The term "fraction of budget" denotes the proportion of allocated budget utilized during training. These results are derived from a scenario where R=150 and the filtering ratio equals 3. Each data point within these plots signifies a round of early stopping. The "fraction of budget" serves as an indicator of the early stopping point, specifically denoting the epoch at which early stopping is executed, calculated based on the filtering ratio ($\eta$) and R. Furthermore, the count of retained model candidates is determined according to the filtering ratio in each early stopping iteration.
>
> We have added explanations in Section 2 and Insight 2.

---

### Official Review · Reviewer_F2DD · 2023-11-01

**Soundness:** 3 good
**Presentation:** 3 good
**Contribution:** 3 good
**Rating:** 8
**Confidence:** 3

**Summary:**

This paper  systematically examines various early stopping criteria in HPO and proposes effective uncertainty-driven criteria .

**Strengths:**

The insights and conclusions look reasonable and well-supported.

Well-organised and quite easy to follow the logic.

**Weaknesses:**

The significant differences among different criteria from section 2.2 would be expected as claimed in the paper;

Including brief introduction about the 9 HPO tasks would be better for readers to understand the problem.

Figure 4 could be improved. I suppose Figure 4(a) and 4(c) are for three tasks of various datasets. The meaning of Figure 4(b), as indicated from the paragraph below, the three curves are for three training settings. If this is the case, I am wondering which task/dataset it is from.

**Questions:**

see above Weaknesses.

---

> ### Author Response · Authors · 2023-11-20
> **Response to Reviewer F2DD**
>
> We sincerely thank Reviewer F2DD for the positive feedback and valuable suggestions. Below we would like to give detailed responses to each of your comments.
>
> **HPO tasks**
>
> Thanks for the suggestion. We have followed your advice to add a description of the 9 HPO tasks and the 2 newly added benchmarks in Appendix A, including their model structures, number of configurations, hyperparameters, loss function, etc.
>
> **Explanation of Fig. 4**
>
> Thanks for the suggestion. Figures 4(a) and (b) show three randomly selected models from the ImageNet task, where 4(a) mainly depicts the fluctuations in validation loss between neighboring epochs, while 4(b) mainly illustrates the model uncertainty contained in different runs. Figure 4(c) shows the relationship between early-stage fluctuations and final test loss for all models in ImageNet, Cifar10, and Cifar100, intended to illustrate that models with large early uncertainty tend to have good final capabilities. We have updated Figure 4 to make it clearer.

---

### Official Review · Reviewer_dT38 · 2023-11-08

**Soundness:** 3 good
**Presentation:** 2 fair
**Contribution:** 1 poor
**Rating:** 3
**Confidence:** 5

**Summary:**

This paper offers an analysis of the implications that arise from using different metrics for decision making in multi-fidelity HPO. It offers several insights, including that sometimes using training performance for decision making is better than using validation performance and also offers some explanations for this and remedies that reduce the risk of using the wrong variables for decision making in pratcice.

**Strengths:**

I think the paper addresses a potentially interesting question and also employs some formal rigor in the analysis. It is mostly clearly written and also coherent in the steps taken.

**Weaknesses:**

I have to admit that I am a bit confused about the contribution and relevance of the paper, and also, in parts, about some of its technical aspects.

Before going into the details, I would like to make a subtle remark on the terminology. The paper consistently talks about early stopping, but there is some recent effort to distinguish this type of stopping (early *discarding*) from simple early stopping, which is the classical case in GradientBoosting or neural networks, i.e., to stop learning as soon as the *learning curve* is stall, independently of performances observed for other learners. This paper uses the term early stopping for both simultaneously. A detailed explanation of this can be found in [1]. The usage of the term "early stopping" is particularly confusing in the context of Hyperband, because literally in the classical sense this has nothing to do with early stopping but re-evaluating (re-training) the model on different budgets. I nthe context of the paper, since neural networks are considered, one can think of Hyperband a bit like a freeze-thaw mechanism, but still this is early discarding and not early stopping, the latter of which would only examine learning curves in isolation.

At a high level, I am missing a clear contribution in this paper and also a red path. The paper reads a bit like a collection of experiments that somebody conducted to study different aspects of uncertainty in HPO but also to analyze certain methods. Most of the insights are very intuitive and would not even require a detailed study. So I guess I also miss some rigor in the paper and do not have a clear takeaway. It is strange, because the paper is not technically poor, and also the topic itself is interesting. Still I feel like I did not learn anything particularly useful reading the paper. Also, the paper is somewhere between an insight paper and proposing certain techniques to improve a situation. For an insight paper, the data basis is too small and the insights are much too vague. For a technical contribution paper, the comparison is lacking. I was very much reminded of one very successful insight paper submitted to this very venue on the effect of the batch size on overfitting [2]. The paper is not related topic-wise but provides a very nice technical explanation on a rather surprising phenomenon, while what we see in this paper is mostly not surprising but very expected. This does not mean that the topic is not interesting, but I think it needs to be exposed in a different way.

here are some detailed remarks on some parts of the paper.


1. I do not really understand Section 2. The authors suggest that the evaluation criterion could be a choice of the HPO tool for early discarding, but this is often not really a choice but a requirement imposed by the use case. While it is true that in NNs one often has both accuracy and loss, it is not really clear to me what the authors here see as the decision variable. Also, I think that "performance metric" or "performance measure" could be a more reasonable term than "criterion", because criteria often suggests also aspects different to performance such as runtime. As a matter of fact, the term "criterion" is used later in the paper to refer to other concepts (Sec. 4). I presume that they want to say something like "it is not clear whether the best metric for stopping is the loss that is being optimized or an external metric such as accuracy and whether one should use the training performance or validation performance". This would then motivate the four cases they look at, but the author's don't phrase it like this, so I am not even sure whether I understood the motivation right.

Next, in the same section, the author's talk about reliability of criteria. What is this now supposed to mean? Implicitly, it gets clear later in 2.2 that the authors refer to test performance as some kind of ground truth and the question is whether the measures used for decision making are faithful in the sense that they do not discard the candidate with the eventually best test performance. At least to me, it is not clear why test accuracy is the eventual objective. This is not necessarily so (accuracy is a non-continuous metric, so one could at least argue for log-loss or Brier score), and in these cases one could also uses these as a loss. Btw. what did the authors use as a loss? I presume its cross-entropy loss, i.e., log-loss, which would make it even less clear why the objective should be test accuracy.

I also do not understand the selection of results for Fig. 1. It seems it is the NAS201 datasets + 2 from LCBench. Is this right? Why do you clip away 4 of the LCBench datasets?

2. In the theoretical part, I see several issues. First, the section lacks a bit clarity on whether D refers to the training data or validation data. Apparently this precisely depends, because D is the data used to estimate \hat f^t, and this can be either training or validation data, depending on whether one computes training/validation accuracy/loss. More importantly, there is a logical mistake. You cannot argue that tighter bounds on the regret imply lower regrets (this is only a conjecture), because the regrets can easily move always in the same range, and only in one case you have higher slacks to the bounds. Personally, I also find the statements a bit contradictory to the observations in Sec. 2, because on the "complex" tasks like CIFAR and ImageNet, not only the training data is bigger but also the validation fold should be much bigger than on the smaller datasets, so one would expect that the validation performance is also more reliable as a criterion. Apparently this is not the case, but I do not currently see how Sec. 3 resolves these doubts.

3. I think that the complexities of the networks in NAS201 and LCBench is inherently very different, which adds a confounding factor to the whole setup. I presume that the variability in NAS201 is much higher than in LCBench, which only has funnel-shaped networks.

4. The experiments conducted in 4.2.1 remain largely unclear to me. I mean, I understand what the authors want to demonstrate, but a lot of technical details are missing so that I it is hard to interpret the results.

[1] https://arxiv.org/abs/2201.12150

[2] https://arxiv.org/abs/1609.04836

**Questions:**

What did you use as the loss in the neural networks? Cross entropy?

In the theoretical part, shouldn't we expect better decision making precisely for large datasets, i.e. isn't this contrary to the findings in Sec. 2? I guess it can next be explained by |D| not being the only factor but also the model complexity, which induces higher variance if |D| is large and the model is flexible. Would you agree?

---

> ### Author Response · Authors · 2023-11-20
> **Response to Reviewer dT38 (Part 1/2)**
>
> We sincerely thank Reviewer dT38 for the valuable suggestions. Below we would like to give detailed responses to each of your comments.
>
>
> **The "early stopping" terminology**
>
> Thanks for pointing out this concern. The term "early stopping" was inherited from the Hyperband paper [1], where it broadly signifies determining when model training should halt based on predefined metrics and strategies. We agree with you that in the context of Hyperband, “early discarding” is a more accurate expression. Yet, we would like to clarify that early discarding represents just one case covered by early stopping. Numerous HPO algorithms, such as Sub-sampling [2], rely on early stopping mechanisms where halted model candidates remain active and might resume training in subsequent stages.
>
> The performance metrics investigated in our study serve to rank model candidates, providing basis for early stopping mechanisms to make decisions. However, whether the early stopping mechanisms entail discarding falls outside of this paper's focus. To validate the broad applicability of our findings, we have introduced experiments on two additional HPO algorithms, BOHB and Sub-sampling, detailed in Appendices A, B, and E.
>
> [1] https://arxiv.org/abs/1603.06560
>
> [2] https://arxiv.org/abs/2209.12499
>
> **Contribution and read path**
>
> Thanks for raising this valuable question. This work belongs to an insight paper, aiming at exploring the impact of diverse performance metrics within the early stopping mechanisms of HPO algorithms. Our experiments centered around commonly used metrics, including training/validation loss/accuracy. The results highlighted the superior efficacy of training loss, especially in intricate tasks with slower convergence. This finding is intriguing, considering that validation loss has been widely adopted. We substantiated this observation through experiments and analysis, identifying factors - model uncertainty and dataset size - being potential reasons for the inferior performance of validation metrics. Furthermore, we explored strategies leveraging uncertainty to enhance the reliability of these metrics.
>
> We have followed your advice to improve the organization of this paper. In particular, we have given a summary of our findings in the Introduction for readers to grasp the message. We also revised some sentences in Section 1 to make our motivation clearer. We've expanded the experimental scope in Appendices A, B, and E, introducing new benchmarks (MLP and LogReg from HPOBench) and incorporating additional HPO algorithms (BOHB and Sub-sampling), to enrich the data basis of this work.
>
> **Decision variable and motivation**
>
> We fully agree with you that in most cases the metrics are determined by user needs. As we pointed out in Section 1, the choice of metrics within HPO is based on the practitioners' personal preferences. Regarding decision variables, our study has shown that delving into the reliability of metrics yields benefits not only for users but also for HPO tools. From the user's perspective, if there is a priori knowledge of a model's convergence behavior, one can choose between training and validation losses based on whether overfitting occurs at early stopping points. Specifically, if there is overfitting, validation loss is preferred; otherwise, it is the training loss. For the HPO tools, they can request both training and validation losses from users, thus incorporating an overfitting detection to avoid unfair comparisons, and making decisions based on which metric contains less uncertainty. Additionally, they can introduce uncertainty, as shown in Section 4, to design more robust metrics for more reliable decision-making. We have outlined general guidelines in Section 5.
>
> Regarding the motivation of this paper, we have followed your suggestion to replace "criterion" with "performance metric" and made a clear distinction in Section 1. In addition to comparing loss and external metrics (e.g., accuracy), we also seek to compare the differences between training and validation metrics. We have modified the expressions in Section 1 and Insight 1 to make the motivation clearer.

---

> ### Author Response · Authors · 2023-11-20
> **Response to Reviewer dT38 (Part 2/2)**
>
> **Final objective**
>
> We use the concept of reliability to assess if a specific performance metric shows statistically significant superiority over others. In our experiments, cross-entropy serves as the loss function, and we choose test accuracy as the final objective based on following considerations:
>
> First, we use balanced accuracy throughout our experiments, except for ImageNet. However, our findings demonstrate that using both test loss and test accuracy yields consistent results. Through an analysis of the relationship between final test loss and accuracy across all models, we unveil a significant linear correlation. The difference in accuracy among models with nearly identical losses (differences around 1e-6) is less than 3%. Conversely, for models achieving the same accuracy, the difference in losses is within 0.4. Notably, in the case of ImageNet, the achieved differences in accuracy and loss stand at 1.7e-7% and 0.08, respectively. Further empirical evidence along with clarifications regarding the balance of benchmark classes can be found in Appendix A.
>
> Second, the test dataset itself might be subject to diverse biases and uncertainties. Consequently, neither test loss nor test accuracy can be deemed as definitive assessment metrics. Cross-entropy loss, sensitive to subtle differences between predicted and true distributions, might respond differently to slight variations in class probabilities. While test loss offers a more nuanced model ranking, it might also contain additional noise or uncertainty. In our comparison scenario, we need only focus on overall significant differences to determine model reliability.
>
> Third, test accuracy, due to its simplicity and intuitive nature, aligns more closely with practical application goals. Real-world scenarios such as image classfication competitions [1] and LLM leadearboards [2] predominantly favor test accuracy as the final objective.
> We have added experiments in Appendix A with test loss as the final objective. The results show that the significance of the differences obtained using test loss and test accuracy as the final objective are generally aligned. These results consistently highlight the superiority of training loss and emphasize the advantage of incorporating uncertainty.
>
> [1] https://paperswithcode.com/sota/image-classification-on-imagenet
>
> [2] https://crfm.stanford.edu/helm/latest/
>
> **Dataset selection in Figure 1**
>
> Because the models in LCBench all converge very quickly, producing overfitted and disordered results, we randomly chose two datasets for illustration. We have added results for the other datasets in Appendix B.1.
>
> **Explanation to the theoretical part**
>
> We fully agree with the comment that D can be a training or validation set, and that model complexity plays a role in HPO efficacy. However, we would like to clarify that our discussion is in the context of "one early-stopping based HPO", which implies a fixed DL task, similar model complexity (with different hyperparameters), and fixed training, validation, and testing sets. Our goal is to allow users or HPO tools to select the best metric when faced with an HPO task, where there is no need to compare between multiple HPO tasks. We have added a detailed description in Section 3 to provide further clarity.
>
> Section 3 aims to delve into strategies for minimizing decision errors, which we define as the probability of the optimal model's loss being greater than the loss of a suboptimal model. By minimizing the probability, it supports the conclusion that "using a larger dataset or choosing more discriminative metrics can yield more reliable results". The training metrics stand out due to their foundation on a larger database, offering more stable values, as demonstrated in our experiments on Nas-Bench-201 and the new benchmarks outlined in Appendix A, B, and E. While validation metrics outperform training metrics on certain datasets like Fashion-MNIST and jasmine, this discrepancy does not contradict our statements. Instead, as explained in the second paragraph of Section 3, it stems from lower model complexity leading to overfitting.
>
> We have refined Insight 1 and highlighted certain statements in Section 3 to enhance the clarity of our findings. We have also updated Section 4.2.1 and Appendices A and E with additional technical details.

---

### Author Response · Authors · 2023-11-20
**General Response: New Results & Paper Updates**

We sincerely thank all the reviewers and the area chair for their efforts in reviewing our paper. Below, we would like to highlight some new results and paper updates during the discussion period.

**New results.**
Following the questions raised by Reviewers dT38 and wZMM, we have conducted experiments on new benchmarks (LogReg and MLP [1]) alongside new HPO algorithms (BOHB [2] and Sub-sampling [3]). The experiments indicate a diminishing gap between metric values as the HPO algorithm is refined. Nevertheless, the outcomes generally affirm the conclusions outlined in the main text, emphasizing the superior performance of training loss due to its extensive data foundation and minimal uncertainty. Additionally, our study emphasizes the efficacy of introducing uncertainty to build robust metrics, thereby enhancing HPO results. Please refer to Appendix A for comprehensive technical details and experimental setups. Comprehensive experimental results are presented in Appendices A, B, and E.

---

**Paper updates.**
Below we highlight the major updates of the revised submission.
- **Section 1**: Add a comprehensive summary of this paper's findings and conclusions for readers to grasp the message. (Reviewers dT38 and wzMM)
- **Section 5**: provide an overall guideline for metric selection. (Reviewer dT38)
- **Appendix A**: provide an introduction to the three HPO algorithms; add an explanation of benchmarks and HPO tasks; add an empirical analysis comparing test loss and test accuracy as final objectives. (Reviewers dT38, F2DD, and wzMM)
- **Appendix B.1**: aggregate all experimental results derived from the Hyperband algorithm; add results for the additional benchmarks in LCBench as a supplementary to Figure 1 in the main text; present results for the new benchmarks (MLP and LogReg). (Reviewers dT38 and wzMM)
- **Appendix B.2**: aggregate all experimental results obtained from the BOHB algorithm, covering Nas-Bench-201, LCBench, MLP, and LogReg. (Reviewers dT38 and wzMM)
- **Appendix B.3**: aggregate all experimental results obtained from the Sub-sampling algorithm, covering Nas-Bench-201, LCBench, MLP, and LogReg. (Reviewers dT38 and wzMM)
- **Appendix E.1**: add technical details for the experiments in Section 4.2.1. (Reviewer dT38)
- **Appendix E.2-E.3**: add results obtained from the BOHB and Sub-sampling algorithms evaluating metrics that integrate uncertainty. (Reviewers dT38 and wzMM)

[1] https://arxiv.org/abs/2109.06716

[2] https://arxiv.org/abs/1807.01774

[3] https://arxiv.org/abs/2209.12499

---

### Meta-Review · Area_Chair_6P5a · 2023-12-06

**Metareview:**

The authors address the problem of hyperparameter optimization (HPO) and specifically focus on the role of early stopping criteria in enhancing HPO efficiency. They systematically explore how the selection of early stopping criteria influences the reliability of decisions and overall outcomes in HPO. The authors propose a set of criteria that incorporate uncertainty and demonstrate their practical importance in improving the reliability of early stopping decisions. Empirical experiments on HPO and NAS benchmarks highlight the critical role of criterion selection and provide insights into the potential benefits of incorporating uncertainty. The research aims to guide the selection and formulation of criteria for a better understanding of mechanisms in early stopping-based HPO.

There is agreement among the reviewers that the paper addresses a relevant, important, and timely topic, and that is contains some potentially interesting insights. The authors efforts to improve the paper during the reviewing phase is also appreciated and acknowledged. That said, the reviewers also believe that the message of the paper is not as clear as it should be. Combined with some terminological issues, this makes it unnecessarily difficult to follow the authors' line of thought. Some theoretical results, in addition to empirical evidence, would further strengthen the paper. In the end, there was a consensus that the paper holds promise and has enough potential, but doesn't yet reach the level expected for this conference.

**Justification For Why Not Higher Score:**

In terms of presentation, the paper doesn't yet reach the level expected for ICLR. It is also lacking in terms of profoundness and depth of its main contributions.

**Justification For Why Not Lower Score:**

N/A

---

### Decision · Program_Chairs · 2024-01-16

Reject